

# High net $CO_2$ and $CH_4$ release at a eutrophic shallow lake on a formerly drained fen

**D. Franz[1], F. Koebsch[1], E. Larmanou[1], J. Augustin[2] and T. Sachs[1]**

[1] {Helmholtz Centre Potsdam, GFZ German Research Centre for Geosciences, Telegrafenberg, 14473 Potsdam, Germany}

[2] {Institute for Landscape Biogeochemistry, Leibniz Centre for Agricultural Landscape Research (ZALF), Eberswalder Str. 84, 15374 Müncheberg, Germany}

*Correspondence to:* D. Franz (daniela.franz@gfz-potsdam.de)

## Abstract

Drained peatlands often act as carbon dioxide ($CO_2$) hotspots. Raising the groundwater table is expected to reduce their $CO_2$ contribution to the atmosphere and revitalize their function as carbon (C) sink in the long term. Without strict water management rewetting often results in partial flooding and the formation of spatially heterogeneous, nutrient-rich shallow lakes. Uncertainties remain as to when the intended effect of rewetting is achieved, as this specific ecosystem type has hardly been investigated in terms of greenhouse gas exchange (GHG) exchange. In most cases, methane ($CH_4$) emissions increase under anoxic conditions due to a higher water table and in terms of global warming potential (GWP) outperform the shift towards $CO_2$ uptake, at least in the short-term.

Based on eddy covariance measurements we studied the ecosystem–atmosphere exchange of $CH_4$ and $CO_2$ (NEE) at a shallow lake situated on a former fen grassland in Northeast (NE) Germany. The lake evolved shortly after flooding, 9 years previous to our investigation period. The ecosystem consists of two main surface types: open water (inhabited by submerged and floating vegetation) and emergent vegetation (particularly including the eulittoral zone of the lake, dominated by *Typha latifolia*). To determine the individual contribution of the two main surface types to the net $CO_2$ and $CH_4$ exchange of the whole lake ecosystem, we combined footprint analysis with $CH_4$ modelling and NEE partitioning.

The $CH_4$ and $CO_2$ dynamics were strikingly different between open water and emergent vegetation. Net $CH_4$ emissions from the open water area were around 4-fold higher than from emergent vegetation stands, accounting for 53 and 13 g $CH_4$ m$^{-2}$ a$^{-1}$, respectively. In addition, both surface types were net



$CO_2$ sources with 158 and 750 g $CO_2$ m$^{-2}$ a$^{-1}$, respectively. Unusual meteorological conditions in terms
of a warm and dry summer and a mild winter might have facilitated high respiration rates. In sum, even
after 9 years of rewetting the lake ecosystem exhibited a considerable C loss and global warming
impact, the latter mainly driven by high $CH_4$ emissions. We assume the eutrophic conditions in
combination with permanent high inundation as major reasons for the unfavourable GHG balance.
**1    Introduction**
Peatland ecosystems play an important role in global greenhouse gas (GHG) cycles, although they
cover only about 3 % of the earth´s surface (Frolking et al. 2011). Peat growth depends on the
proportion of carbon (C) sequestration and release. Pristine peatlands act as long-term C sinks and are
near-neutral (slightly cooling) regarding their global warming potential (GWP; Frolking et al. 2011),
dependent on rates of C sequestration and methane ($CH_4$) emissions. However, many peatlands
worldwide are used e.g. for agriculture, as are more than 85% of the peatlands in Germany and the
Netherlands (Silvius et al. 2008). Drainage is associated with shrinkage and internal phosphor
fertilisation of the peat (Zak et al. 2008). Moreover, the hydrology of the area as well as physical and
chemical peat characteristics are changing (Holden et al. 2004, Zak et al. 2008). Above all, drained and
intensively managed  peatlands are known as strong sources of carbon dioxide ($CO_2$; e.g. Joosten et al.
2010, Hatala et al. 2012, Beetz et al. 2013). On the other hand, lowering the water table is typically
accompanied with decreasing $CH_4$ emissions (Roulet et al. 1993). Emission factors of 1.6 g $CH_4$ m$^{-2}$ a$^{-1}$
and 2235 g $CO_2$ m$^{-2}$ a$^{-1}$ were assigned to temperate deep-drained nutrient-rich grassland in the 2013
wetland supplement to the 2006 IPCC Guidelines for National Greenhouse Gas Inventories (IPCC 2014).
In the last decades rewetting of peatlands attracted attention in order to stop soil degradation, reduce
$CO_2$ emissions and to recover their functions as C and nutrient sink and ecological habitat (Zak et al.
2015). Large rewetting projects were initiated, e.g. the Mire Restoration Program of the federal state
of Mecklenburg-West Pomerania in Northeast (NE) Germany (Berg et al. 2000) starting in 2000 and
involving 20 000 ha of formerly drained peatlands, thereby especially fens (Zerbe et al. 2013) e.g. in
the Peene river catchment. However, uncertainties remain as to when the intended effects of
rewetting are achieved. Only few studies exist on the temporal development of GHG emissions of
rewetted fens, especially on longer time scales. Augustin and Joosten (2007) discuss three very
different states following peatland rewetting based on observations at Belarusian mires, though
without specifying the individual lengths of the phases. Broad agreement exists concerning the $CH_4$
hot spot characteristic of newly rewetted peatlands (e.g. Meyer et al. 2001, Hahn-Schöfl et al. 2011,



Knox et al. 2015). However, a rapid recovery of the net $CO_2$ sink function is not consistently reported
(e.g. Wilson et al. 2007).
Peatlands develop a pronounced microtopography after drainage and subsequent subsidence.
Rewetting e.g. in the Peene river catchment resulted in the formation of large-scale shallow lakes in
the lower parts of the fens, with water depths usually below 1 m (Zak et al. 2015, Steffenhagen et al.
2012). These new ecosystems are nutrient-rich and most often strikingly different from natural
peatlands. They experience a rapid secondary plant succession (Zak et al. 2015). Helophytes are
expected to progressively enter the open water body over the time leading to the terrestrialisation of
the shallow lake and in the best case peat formation. However, this new ecosystem type and its
progressive transformation have hardly been investigated in terms of GHG dynamics. The ecosystem-
inherent spatial heterogeneity suggests complex patterns of GHG emissions due to distinct GHG source
or sink characteristics of the involved surface types (generally open water and the littoral zone)
resulting in measurement challenges. Site-specific heterogeneity implicitly has to be considered for
the evaluation of ecosystem scale flux measurements (e.g. Barcza et al. 2009, Hendriks et al. 2010,
Hatala Matthes et al. 2014). The importance of small open water bodies in wetlands as considerable
GHG sources was highlighted in previous studies (e.g. by Schrier-Uijl et al. 2011, Zhu et al. 2012, IPCC
2014) and in case of $CH_4$ even for landscape-scale budgets e.g. by Repo et al. (2007). In addition, the
littoral zone of lakes is often found to be a $CH_4$ hot spot (Juutinen et al. 2003, Wang et al. 2006) with a
contribution of up to 90 % to the whole-lake $CH_4$ release (Smith and Lewis 1992), albeit depending on
the lake size (Bastviken et al. 2004) and plant community. Rõõm et al. (2014) measured the largest $CH_4$
(and $CO_2$) emissions of a temperate eutrophic lake at the helophyte zone within the littoral.
The objectives of this study are 1) to investigate the ecosystem-atmosphere exchange of $CH_4$ and $CO_2$
(NEE) of a nutrient-rich lake ecosystem emerged at a former fen grassland and 2) particularly infer the
individual GHG dynamics of the main surface types within the ecosystem and quantify their
contribution to the annual exchange rates. Therefore, we applied the eddy covariance technique from
May 2013 to May 2014 and used an analytical footprint model to downscale the spatially integrated,
half-hourly fluxes to the main surface types "open water" and "emergent vegetation". The resulting
source area (i.e. spatial origin of the flux) fractions were then included in a temperature response ($CH_4$)
and NEE partitioning model ($CO_2$) in order to quantify the source strength of the two surface types.



## 2 Material and methods

### 2.1 Study site

The study site "Polder Zarnekow" is a rewetted, rich fen (minerotrophic peatland) located in the Peene river valley (Mecklenburg-West Pomerania, NE Germany, 53°52.5' N 12°53.3' E, see Fig. 1), with less than 0.5 m a.s.l. elevation. It is part of the Terrestrial Environmental Observatories Network (TERENO). The temperate climate is characterised by a long-term mean annual air temperature and mean annual precipitation of 8.7 °C and 584 mm, respectively (German Weather Service, meteorological station Teterow, 24 km SW of the study site; reference period 1981–2010). The geomorphological character of the area is predominantly a result of the Weichselian glaciation as the last period of the Pleistocene (Steffenhagen et al. 2012). The fen developed with continuous percolating groundwater flow (Succow 2001). Peat depth partially reaches 10 m (Hahn-Schöfl et al. 2011). Drainage was initialized in the 18$^{th}$ century and strongly intensified between 1960 and 1990 within an extensive melioration program (Höper et al. 2008). The decline of the water table to > 1 m below surface and subsequent decomposition and mineralisation of the peat (especially in the upper 30 cm, Hahn-Schöfl et al. 2011) caused phosphor fertilisation (Zak et al. 2008) and soil subsidence to levels below that of adjacent freshwater bodies (Steffenhagen et al. 2012, Zerbe et al. 2013). The latter simplified the rewetting process which was initiated in winter 2004/2005 by opening the dikes.

In consequence of flooding the drained fen was converted into a spatially heterogeneous site of emergent vegetation (on temporarily inundated soil) and permanent open water areas. In this study we focus on a eutrophic shallow lake (open water body about 7.5 ha) as part of the rewetted area, with water depths ranging from 0.1 to 0.7 m. During the study period the open water body of the lake was inhabited by submerged and floating macrophytes, particularly *Ceratophyllum demersum*, *Lemna minor*, *Spirodela polyrhiza* (Steffenhagen et al. 2012) and *Polygonum amphibium*, which rather corresponds to the sublittoral zone in a typical lake zonation. *Ceratophyllum* and *Lemna* sp. were already reported to colonise the lake in the second year of rewetting (Hahn-Schöfl et al. 2011). *Phalaris arundinacea*, that dominated the fen before rewetting, died off in the first year of inundation (Hahn-Schöfl et al. 2011) and has been limited to the non-inundated periphery of the ecosystem. Helophytes (e.g. *Glyceria*, *Typha*) started the colonisation of lake margins and other temporarily inundated areas in the third year of rewetting. The eulittoral zone of the lake is now dominated by *Typha latifolia* stands gradually colonising the open water in the last years. Emergent vegetation stands also include sedges as *Carex gracilis* (Steffenhagen et al. 2012). At the bottom of the shallow lake an up to 30 cm thick layer of organic sediment evolved, initially fed by fresh plant material of the former vegetation and



since then continuously replenished by recent aquatic plants and helophytes after die-back (Hahn-
Schöfl et al. 2011).
## 2.2   Eddy covariance and additional measurements
We conducted eddy covariance (EC) measurements of $CO_2$ and $CH_4$ exchange on a tower placed on a
stationary platform at the NE edge of the shallow lake (see Fig. 1). Thereby we ensured to frequently
catch the signal from both the open water body and the *Typha latifolia* dominated belt of the shallow
lake (eulittoral zone). We defined an area of interest (AOI) in order to focus on an ecosystem
dominated by a shallow lake and to avoid a possible impact of the farm and grassland to the north of
the shallow lake. The EC measurement setup included: an ultrasonic anemometer for the 3D wind
vector ($u$, $v$, $w$) and sonic temperature (HS-50, Gill, Lymington, Hampshire, UK), an enclosed-path
infrared gas analyser (IRGA) and an open-path IRGA for $CO_2/H_2O$ and $CH_4$ concentrations, respectively
(LI-7200 and LI-7700, LI-COR Biogeosciences, Lincoln NE, USA). Flowrate was about 10-11 l $min^{-1}$.
Measurement height was on average 2.63 m above the water surface at the position of the tower,
depending on the water level. We recorded raw turbulence and concentration data with a LI-7550
digital data logger system (LI-COR Biogeosciences, Lincoln NE, USA) at 20 Hz in half-hourly files. The
dataset is shown in Coordinated Universal Time (UTC), which is 1 hour behind local time (LT).
We further equipped the tower with instrumentation for net radiation, air temperature/humidity, 2D
wind direction and speed, incoming and reflected photosynthetic photon flux density (PPFD/PPFDr)
and water level. Additional measurements in close proximity to the tower included precipitation, soil
heat flux as well as soil and water temperature. Soil temperature was measured below the water
column in depths of 10 cm, 20 cm, 30 cm, 40 cm and 50 cm and water temperature at the sediment-
water-interface.  All non-eddy covariance-related measurements were logged as 1 min averages/sums
(precipitation). Gaps were filled with measurements of the Leibniz Centre for Agricultural Landscape
Research (ZALF, Müncheberg, Germany) at the same platform and  a nearby climate station (Climate
station Karlshof, GFZ German Research Centre for Geosciences, 14 km distance from study site, Itzerott

149  2015).

A water density gradient was calculated based on the temperature at the water surface and at the
sediment-water interface. The water surface temperature was calculated based on the Stefan-
Boltzmann law (see e.g. Foken et al. 2008):
$$T_w = \sqrt[4]{\frac{I}{\varepsilon_w\,\sigma_{SB}}}$$   (1)



where $T_w$ is the water surface temperature (K), $I$ is the long-wave outgoing radiation (W m$^{-2}$), $\varepsilon_w$ is
the infrared emissivity of water (0.960) and $\sigma_{SB}$ is the Stefan–Boltzmann constant (5.67·10$^{-8}$ W m$^{-2}$ K$^{-}$
$^4$). We calculated the density of the air-saturated water at the water surface and the sediment-water
interface according to Bignell (1983):
$$\rho_{as} = \rho_{af} - 0.004612 + 0.000106 * T \qquad (2)$$
where $\rho_{as}$ is the density of the respective air-saturated water (k m$^{-3}$), $\rho_{af}$ is the density of the
respective air-free water (k m$^{-3}$; see Wagner and Pruß 2002) at atmospheric pressure (1013 hPa) and
$T$ is the respective water temperature (°C). The gradient of the two water densities (air-saturated)
$\Delta\rho/\Delta z$ was calculated as difference of the water density (air-saturated) at the sediment-water
interface and the surface water density (air-saturated), divided by the distance (m) between the two
basic temperature measurements. Changes of the distance due to the fluctuating water level were
considered. Positive and negative gradients indicate periods of stratification and thermally induced
convective mixing of the water column, respectively.
## 2.3   Flux computation and further processing
For this analysis we used data from 14 May 2013 to 14 May 2014. We calculated half-hourly fluxes of
$CO_2$ and $CH_4$ based on the covariances between the respective scalar concentration and the vertical
wind velocity using the processing package EddyPro 5.2.0 (LI-COR, Lincoln, Nebraska, USA). Sonic
temperature was corrected for humidity effects according to van Dijk et al. (2004). Artificial data spikes
were removed from the 20 Hz data following Vickers and Mahrt 1997. We used the planar fit method
(Finnigan et al. 2003, Wilczak et al. 2001) for axis rotation and defined the sector borders according to
Siebicke et al. (2012). Block averaging was used to detrend turbulent fluctuations. For time lag
compensation we applied covariance maximization (Fan et al. 1990). Spectral losses due to crosswind
and vertical instrument separation were corrected according to Horst and Lenschow (2009). The
methods of Moncrieff et al. (2004) and Fratini et al. (2012) were used for the correction of high-pass
filtering and low-pass filtering effects, respectively. For fluctuations of $CH_4$ density we corrected
changes in air density according to Webb et al. (1980), considering LI-7700-specific spectroscopic
effects (McDermitt et al. 2011). According to the micrometeorological sign convention, positive values
represent fluxes from the ecosystem into the atmosphere (emission) and negative values fluxes from
the atmosphere into the ecosystem (ecosystem uptake).





### 2.4 Quality assurance


We filtered the averaged fluxes according to their quality as follows (see Table 1):
- We rejected fluxes with quality flag 2 (QC 2, bad quality) based on the 0-1-2 system of Mauder
and Foken (2004).
- $CH_4$ fluxes were skipped if the signal strength (RSSI) was below the threshold of 14 %. This
threshold was estimated according to Dengel et al. (2011).
- Fluxes with friction velocity ($u^*$) < 0.12 m s$^{-1}$ and > 0.76 m s$^{-1}$ were not included due to
considerably high fluxes beyond these thresholds, which were estimated similar to the
procedure described in Aubinet et al. (2012) based on binned u$^*$ classes. The storage term was
calculated as described in Béziat et al. (2009).
- Unreasonably high positive and negative fluxes (0.2 %/99.8 % percentile) were discarded from
the $CO_2$ and $CH_4$ flux dataset.
Quality control (apart from EddyPro internal steps) and the subsequent processing steps were
performed with the free software environment R (R Core Team 2012).

### 2.5 Footprint modelling


We applied footprint analysis to determine the source area including the fractions of the surface types
of each quality-controlled half-hourly flux using a footprint calculation procedure following Göckede
et al. (2004). The source area functions were calculated based on the analytical footprint model of
Kormann and Meixner (2001). Roughness length and vegetation height were estimated with an
iterative algorithm (see also Barcza et al. 2009). Based on an aerial image (GoogleEarth,
http://earth.google.com/) the surface of our study site was classified into two main types and
implemented in a land cover grid: "open water" including in particular the open waterbody of the
shallow lake with 0.1 to 0.7 m water depth and "emergent vegetation" with a height up to 2 m and
including the eulittoral zone of the shallow lake dominated by *Typha latifolia*. The cumulative annual
footprint climatology was calculated following Chen et al. (2011). Fluxes were excluded where
footprint information was not available or more than 20 % of the source area was outside the AOI (see
Fig. 1 and Table 1). The fractional coverage within the AOI ($A_i$) was 21.7 % for open water.
Quasi-continuous source area information for the two surface types were achieved by gapfilling the
results of the footprint model with the means of the source area fractions of the surface types ($\Omega_i$) for
1°-wind direction-intervals, separately for stable and unstable conditions. In case the sum of the $\Omega_i$ was





less than 100 %, when the source area exceeded the set borders, we assigned the remaining
contribution percentages to emergent vegetation, as the area beyond the borders is dominated by
emergent vegetation rather than open water.
**2.6 Gapfilling**
An enhanced lookup table (LUT) approach proposed by Reichstein et al. (2005), available as web tool
based on the R package REddyProc (http://www.bgc- jena.mpg.de/REddyProc/brew/REddyProc.rhtml)
was applied for gapfilling and partitioning of NEE measurements (LUT$_{CO2nofoot}$), with air temperature as
temperature variable. For the gapfilling of $CH_4$ measurements non-linear regression (NLR) was applied
(NLR$_{CH4nofoot}$):
$$F_{CH_4} = \exp(a + b_1 \cdot X_1 + \ldots + b_j \cdot X_j) \qquad (3)$$
where $a$ and $b_1 \ldots b_j$ are fitting parameters and $X_1 \ldots X_j$ are environmental parameters. Several
environmental parameters, which were reported to be correlated with $CH_4$ flux on different time
scales, were tested to find the best bi- or multivariate NLR model for the ecosystem $CH_4$ flux: pressure
change, u*, PAR, air temperature, soil heat flux, soil/peat temperature in different heights and
waterlevel. Only fluxes of the best quality (QC 0) were used to fit the NLR model and the LUT.
**2.7 Calculation of the annual CO₂ and CH₄ budget and the global warming**
**potential (GWP)**
We used the continuous flux datasets derived from gapfilling for the calculation of annual $CO_2$ and $CH_4$
budgets. The ecosystem GHG balance was calculated by summation of the net ecosystem exchange of
$CO_2$ and $CH_4$ using the global warming potential (GWP) of each gas at the 100-year time horizon (IPCC,
2013). According to the IPCC AR5 (IPCC, 2013) $CH_4$ has a 28-fold global warming potential compared
to $CO_2$ (without inclusion of climate-carbon feedbacks).
The uncertainty of the annual estimates was calculated as the square root of the sum of the squared
random error (measurement uncertainty) and gapfilling error within the one-year observation period
(see e.g. Hommeltenberg et al. 2014, Shoemaker et al. 2015). An estimation of the random uncertainty
due to the stochastic nature of turbulent sampling according to Finkelstein and Sims (2001) is
implemented in EddyPro 5.2.0. In case of the LUT approach the gapfilling error (standard error) was
calculated from the standard deviation of the fluxes used for gapfilling, provided by the web tool. For
budgets based on the NLR approach we used the residual standard error of the NLR model as gapfilling
error (following Shoemaker et al. 2001).



## 2.8 Estimation of surface type fluxes


To estimate the specific surface type fluxes, we combined footprint analysis with NEE partitioning
(using NLR) to assign gross primary production (GPP) and ecosystem respiration ($R_{eco}$) to the two main
surface types ($NLR_{CO2foot}$). $R_{eco}$ and GPP were modelled as sum of the two surface type fluxes weighted
by $\Omega_i$ (analogous to Forbrich et al. 2011). Night-time $R_{eco}$ (global radiation < 10 W m$^{-2}$) was estimated
by the exponential temperature response model of Lloyd and Taylor (1994) assuming that night-time
NEE represents the night time $R_{eco}$:
$$R_{eco} = \sum_{i=1}^{2} \Omega_i \cdot (R_{ref_i} \cdot \exp(E_{0_i}(\frac{1}{T_{ref}-T_0} - \frac{1}{T_{air}-T_0}))) \tag{4}$$
where $R_{eco}$ is the half-hourly measured ecosystem respiration (µmol$^{-1}$m$^{-2}$s$^{-1}$), $\Omega_i$ is the source area
fraction of the respective surface type, $R_{ref}$ is the respiration rate at the reference temperature $T_{ref}$
(283.15 K), $E_0$ defines the temperature sensitivity, $T_0$ is the starting temperature constant (227.13 K)
and $T_{air}$ the mean air temperature during the flux measurement. The model parameters achieved for
night time $R_{eco}$ were applied for the modelling of day-time $R_{eco}$. GPP was calculated by subtracting
daytime $R_{eco}$ from the measured NEE. GPP was further modelled using a rectangular, hyperbolic light
response equation based on the Michaelis–Menten kinetic (see e.g. Falge et al. 2001):
$$GPP = \sum_{i=1}^{2} \Omega_i \cdot (\frac{GP_{max_i} \cdot \alpha_i \cdot PAR}{\alpha_i \cdot PAR + GP_{max_i}}) \tag{5}$$
where $GPP$ is the calculated gross primary production (µmol$^{-1}$m$^{-2}$s$^{-1}$), $\Omega_i$ is the source area fraction
of the respective surface type, $GP_{max}$ is the maximum C fixation rate at infinite photon flux density of
the photosynthetic active radiation $PAR$ (µmol$^{-1}$m$^{-2}$s$^{-1}$), $\alpha$ is the light use efficiency (mol $CO_2$ mol$^{-1}$
photons). We calculated one parameter set for $R_{eco}$ and GPP per day based on a moving window of 28
days (method $NLR_{nofoot}$). In order to avoid over-parameterization we introduced fixed values of 150 for
$E_0$ and -0.03 and -0.01 for $\alpha$ of emergent vegetation and water bodies, respectively, to get reasonable
parameter values for $R_{ref}$ and $GP_{max}$. We excluded parameter sets for $R_{eco}$ or GPP, if one of the two $R_{ref}$
and $GP_{max}$ parameter values was insignificant (p-value ≥ 0.05), negative or zero. In addition, the 1 %/99
% percentiles of $GP_{max}$ were excluded. These gaps within the parameter set were filled by linear
interpolation. Gaps remain in $R_{eco}$ and GPP time series due to gaps in the environmental variables. Gaps
up to 3 hours length were filled by linear interpolation. Larger gaps were filled with the mean of the
flux during the same time of the day before and after the gap. Due to the moving window approach,
we could not estimate model parameters for the first and last 14 days of our study period. Instead, we
applied the first and last estimated parameter set, respectively. Modelled GPP and $R_{eco}$ were summed
up to half-hourly NEE fluxes and used for alternative NEE gapfilling.





As for NEE we expect different $CH_4$ emission rates of the two surface types. Thus, we extended the NLR
model ($NLR_{CH4nofoot}$) in a way that the $CH_4$ flux is the sum of the two surface type fluxes weighted by $\Omega_i$
($NLR_{CH4foot}$):
$$F_{CH_4} = \sum_{i=1}^{2} \Omega_i \cdot \exp(a_i + b_{1i} \cdot X_1 + \ldots + b_{ji} \cdot X_j) \qquad (6)$$
where $\Omega_1$ is the source area fraction of the respective surface type. Considering the principle of
parsimony, we combined up to three parameters besides the contribution of the surface types.
Remaining gaps were filled by interpolation. Surface type $CO_2$ and $CH_4$ fluxes were derived based on
the fitted NLR parameters.
We calculated the annual budgets of $CO_2$ and $CH_4$ for the EC source area, the surface types (assuming
source area fraction of 100 % for the respective surface type) and the AOI, the latter following Forbrich
et al. (2011) by applying Eq. 4 and Eq. 5 for $CO_2$ as well as Eq. 6 for $CH_4$ with the fitted parameters, but
$A_i$ instead of $\Omega_i$ as weighting surface type contribution. The gapfilling error for the $NLR_{CO2foot}$ model was
based on the residual standard error of both $R_{eco}$ and GPP.

**3   Results**
**3.1   Environmental conditions and fluxes of $CO_2$ and $CH_4$**
Mean annual air temperature and annual precipitation for the study period were 10.1 °C and 416.5
mm, respectively, indicating an unusual dry and warm measurement period compared to the long-
term average. The summer 2013 was among the 10 warmest since the beginning of the measurements
in 1881 (German Weather Service DWD). From June to August monthly averaged air temperature was
0.2 up to 0.9 °C higher and precipitation was 9.1 up to 38.1 mm less than the long-term averages. The
open water area of the shallow lake was densely vegetated with submerged and floating macrophytes.
A summertime algae slick accumulated in the NE part of the shallow lake. Winter 2013/2014 was
characterised by exceptionally mild temperatures and very sparse precipitation. However, a short cold
period (see Fig. 2) resulted in ice cover on the shallow lake between 21 January and 16 February 2014.
The water level of the shallow lake fluctuated between 0.36 and 0.77 m (at the position of the sensor)
and had its minimum at the end of August/beginning of September and its maximum in January. We
observed the exposure of normally inundated soil surface at emergent vegetation stands during the
dry period in summer 2013.
Both $CO_2$ and $CH_4$ flux measurement time series showed a clear seasonal trend with median $CO_2$ flux
of 0.57 $\mu$mol m$^{-2}$ s$^{-1}$ and a median $CH_4$ flux of 0.02 $\mu$mol m$^{-2}$ s$^{-1}$. $CH_4$ emissions peaked in mid-August



2013 with 0.57 $\mu$mol m$^{-2}$ s$^{-1}$. The highest net $CO_2$ uptake (-15.34 $\mu$mol m$^{-2}$ s$^{-1}$) and release (21.04 $\mu$mol
m$^{-2}$ s$^{-1}$) were both observed in June 2013. A diurnal cycle of $CO_2$ fluxes with peak uptake around midday
and peak release around midnight was obvious until November 2013 and beginning in March 2014
(see Fig. 3). To investigate the potential presence of a diurnal cycle of $CH_4$ fluxes we normalized the
mean half-hourly $CH_4$ fluxes per month with the respective median of the half-hourly fluxes of the
specific month (minimum five 30 min fluxes per day; method modified from Rinne et al. 2007). We
found a clear diurnal cycle of $CH_4$ fluxes from June to September 2013 and starting again in March 2014
(April and May not shown as the sensor was dismantled) with daily peaks during night time (around
midnight until early morning). The water density gradient indicates thermally induced convective
mixing of the whole water column during the night (around midnight until early morning) from May
until October 2013 and from February to May 2014. In May 2014 the diurnal pattern of the water
density gradient was less pronounced than in May 2013.
## 3.2   Gapfilling performance and annual budgeting of $CO_2$, $CH_4$, C and GWP
The LUT$_{CO2nofoot}$ approach explained 74 % of the variance in NEE (see Table 2). Median NEE accounted
for 1.9 g $CO_2$ m$^{-2}$ d$^{-1}$. The annual budget of gapfilled NEE (LUT$_{CO2nofoot}$) between 14 May 2013 and 14
May 2014 was 524.5 ± 5.6 g $CO_2$ m$^{-2}$ (see Table 3), characterising the site as strong $CO_2$ source with
moderate rates of R$_{eco}$ and GPP. We found a surprising $CO_2$ release strength during summer 2013,
where already at the end of June daily R$_{eco}$ often exceeded GPP. The highest $CO_2$ emission and uptake
rates of 24.8 g $CO_2$ m$^{-2}$ d$^{-1}$ and -27.9 g $CO_2$ m$^{-2}$ d$^{-1}$ were both observed in the beginning of July 2013 (see
Fig. 2). July 2013 accounted for 23.2 % and 25.8 % of the annual R$_{eco}$ and GPP, respectively. In addition,
net $CO_2$ release outside the growing season (definition of the growing season following Lund et al.
2010; until 19 November 2013 and starting 26 February 2014) was 203.7 g $CO_2$ m$^{-2}$ with a median of
2.2 g $CO_2$ m$^{-2}$ d$^{-1}$.
The environmental variable giving the best NLR model for $CH_4$ was soil temperature in 10 cm depth
(T$_{s10}$):
$$F_{CH_4} = \exp(-7.224 + 0.313 \cdot T_{s10}) \tag{7}$$
The model described 79 % of the variance in $CH_4$ flux (see Table 2). Including additional environmental
variables to the regression function did not increase the model performance significantly. Cumulative
$CH_4$ emissions were 40.5 ± 0.2 g $CH_4$ m$^{-2}$ a$^{-1}$ (see Table 3). Median $CH_4$ emissions were 41.9 mg m$^{-2}$ d$^{-1}$,
peaked at the end of July 2013 with 0.6415 g $CH_4$ m$^{-2}$ d$^{-1}$ and were at the minimum in January 2014
(see Fig. 2). The month with the highest proportion of annual $CH_4$ emissions was August 2013 (27.3 %).





Non-growing season $CH_4$ fluxes only accounted for a small proportion within the annual budget, about
0.8 g $CH_4$ $m^{-2}$.
The site was an effective C and GHG source, accounting for 173.4 ± 1.7 g C $m^{-2}$ $a^{-1}$ and 1658.5 ± 11.2 g
$CO_2$-Eq. $m^{-2}$ $a^{-1}$ for the EC source area (see Fig. 4). The proportion of $CO_2$ in the C and GWP budget was
82.5 % and 31.6 %, respectively. Components of the annual net C balance other than $CO_2$ and $CH_4$
fluxes, e.g. dissolved C, are not considered in this study. Our uncertainty estimates are within the range
of similar studies (e.g. Shoemaker et al. 2015).

## 3.3 Source area composition and spatial heterogeneity of $CO_2$ and $CH_4$ exchange

Footprint analysis revealed the peak contribution in an average distance of 18 m from the tower and
mainly from the open water area of the shallow lake (see Fig. 5). Open water covered on average 62.5
% of the EC source area. The two surface types showed different emission rates in terms of higher $CH_4$
fluxes and lower NEE rates with increasing $\Omega_{water}$ (see Fig. 6). Within the $NLR_{CO2foot}$ approach both
surface types were denoted as sources of $CO_2$, but with about 4-fold stronger rates of GPP, $R_{eco}$ and
NEE for emergent vegetation compared to open water (see Fig. 7 and Table 3). The approach yielded
a similar cumulative annual NEE for the whole EC source area including both surface types as the
$LUT_{CO2nofoot}$ approach, but lower component fluxes (GPP and $R_{eco}$). As for $CO_2$, we implemented $\Omega_i$ as
weighting factors within the NLR model for $CH_4$ ($NLR_{CH4foot}$) to get the surface type specific fluxes of $CH_4$
and fitted the parameters as follows:
$$F_{CH_4} = \Omega_{veg} \cdot \exp(-10.076 + 0.415 \cdot T_{s10}) + \Omega_{water} \cdot \exp(-6.449 + 0.286 \cdot T_{s10}) \quad (8)$$
Open water accounted for more than 4-fold higher emissions than the vegetated areas (see Fig. 7 and
Table 3). The $NLR_{CH4foot}$ approach revealed a similar annual $CH_4$ budget as the $NLR_{CH4nofoot}$ approach.
Annual budgets of $CO_2$ (844 g $CO_2$ $m^{-2}$ $a^{-1}$) and $CH_4$ (22 g $CH_4$ $m^{-2}$ $a^{-1}$) for the AOI differed strongly from
the budgets for the EC source area due to the contrasting emission rates of open water and emergent
vegetation (see Table 3) and different fractional coverages of the surface types within the AOI and the
EC source area. This resulted in a higher C loss (246.5 g C $m^{-2}$ $a^{-1}$) and a lower GWP (1452.9 g $CO_2$-Eq.
$m^{-2}$ $a^{-1}$) than for the EC source area. In the following we will primarily discuss the budgets of the EC
source area and the surface types.



## 4   Discussion

### 4.1   Diurnal variability of CH₄ emissions

In terms of its daily cycle, $CH_4$ exchange between wetland ecosystems and the atmosphere is not generalisable, but rather dependent on the spatial characteristics of the wetland and thus, the impact of the individual $CH_4$ emission pathways (diffusion, ebullition, plant-mediated transport). Our measurements showed a diurnal cycle of $CH_4$ exchange from June to September 2013 and in March 2014, with the strongest emissions during night, as reported for shallow lakes (e.g. Podgrasjek et al. 2014) and wetland sites with a considerable fraction of open water (e.g. Godwin et al. 2013, Koebsch et al. 2015). In comparison, wetland $CH_4$ emissions were also reported to show daily maxima at day-time (e.g. Morrisey et al. 1993, Hendriks et al. 2010, Hatala Matthes et al. 2014), especially at sites with high abundance of vascular plants. No diurnal pattern (e.g. Rinne et al. 2007, Forbrich et al. 2011, Herbst et al. 2011) occurred especially at sites without large open water areas (Godwin et al. 2013).

We assume the process of convective mixing of the water column (e.g. Godwin et al. 2013, Poindexter and Variano 2013, Podgrajsek et al. 2014, Sahlée et al. 2014, Koebsch et al. 2015) to be crucial for the diurnal pattern of $CH_4$ emissions at our study site. This is indicated by the concurrent timing of convective mixing and daily peak $CH_4$ emissions and a generally high fractional source area coverage of the open water, which shows higher rates of $CH_4$ release than emergent vegetation. Furthermore, closed chamber measurements likewise show night-time peak emissions on the shallow lake in summer 2013 (Hoffmann et al. 2015). During the day, $CH_4$ is trapped in the lower (anoxic) layers of the thermally stratified water column. Due to the heat release of the surface water to the atmosphere in the night the surface water cools down, initiating convective mixing of the water column down to the bottom. Diffusion is enhanced due to the buoyancy-induced turbulence, the associated increased gas transfer velocity at the air-water interface (Eugster et al. 2003, MacIntyre et al. 2010, Podgrajsek et al. 2014) as well as the transport of $CH_4$ enriched bottom water to the surface (Godwin et al. 2013, Podgrajsek et al. 2014). In addition, ebullition can be triggered by turbulence due to convective mixing (Podgrajsek et al. 2014, Read et al. 2012). The daily pattern of the open water $CH_4$ release might superimpose the reverse diurnal cycle of plant-mediated transport with peak emissions during day-time, as the release of methane is dependent on the stomatal conductance of the plants (e.g. Morrisey et al. 1993). This pathway is limited to plants with aerenchymatic tissue like *Typha latifolia*, which dominates the eulittoral zone at our study site. $CH_4$ is transported from the soil to the atmosphere, bypassing potential oxidation zones above the rhizosphere (chimney effect). Unusually for wetland



plants (Torn and Chapin 1993), complete stomatal closure during night was observed for *Typha latifolia*
(Chanton et al. 1993).

### 399     4.2   Annual CH$_4$ emissions

The CH$_4$ emissions of our studied ecosystem were within the range of other temperate fen sites
rewetted for several years (up to 63 g CH$_4$ m$^{-2}$ a$^{-1}$; e.g. Hendriks et al. 2007, Wilson et al. 2008, Günther
et al. 2013, Schrier-Uijl et al. 2014). This rate corresponds to twice the emission factor of 21.6 g CH$_4$ m$^{-}$
$^2$ a$^{-1}$, that was assigned to rewetted temperate rich organic soils, which is in turn more than twice the
rate of the nutrient-poor complement (IPCC 2014). In contrast, newly rewetted fens emit its multiple.
In the first year after flooding, Hahn et al. (2015) observed an average net release of 260 g CH$_4$ m$^{-2}$ a$^{-1}$,
which is 186 times higher than before flooding, at a fen site in NE Germany. Two years later the CH$_4$
emissions were significantly lower (40 g CH$_4$ m$^{-2}$ per growing season; Koebsch et al. 2015). However,
natural fens release most often less CH$_4$ than the majority of rewetted fens (e.g. Bubier et al. 1993,
Nilsson et al. 2001), with some exceptions (e.g. Huttunen et al. 2003).
The two main surface types open water and emergent vegetation differed substantially in their CH$_4$
exchange rates. Open water contributed overproportionally to the measured ecosystem fluxes and
showed higher CH$_4$ release rates (52.6 g CH$_4$ m$^{-2}$ a$^{-1}$) than the emergent vegetation stands (13.2 g CH$_4$
m$^{-2}$ a$^{-1}$). However, closed-chamber measurements at the shallow lake show an even higher long-term
average annual CH$_4$ release rate (206 g CH$_4$ m$^{-2}$ a$^{-1}$) since rewetting with large interannual variability
and occasionally extreme high release rates (up to 400 g CH$_4$ m$^{-2}$ a$^{-}$1; Casares et al., in prep.).
We assume the permanent high inundation and high productivity due to eutrophic conditions, feeding
the organic mud deposited at the bottom of the open water body (which is typically for shallow lakes
in rewetted fens), to be of particular importance for high CH$_4$ emissions as substrate for
decomposition. The mud initially evolved as a mixture of sand and easily decomposable labile plant
litter from reed canary grass, which died-off after flooding and produced a large C pool for CH$_4$
production (Hahn-Schöfl et al 2011). During an incubation experiment with substrate from our study
site Hahn-Schöfl et al. (2011) observed that the new sediment layer has very high specific rates of
anaerobic CH$_4$ (and CO$_2$) production. In addition, Zak et al. (2015) emphasised the impact of litter
quality and reported a very high CH$_4$ production potential for litter of *Ceratophyllum demersum,* which
dominates the biomass in the open water at our study site. Due to the eutrophic character of the lake
and associated high productivity within the open water body and in the eulittoral zone, high amounts
of fresh labile organic matter continuously replenish the mud layer and thus the C pool. As the C
balance (CO$_2$ and CH$_4$) seems to be extremely unbalanced, we further assume lateral input of





allochthonous organic matter into the NE "bay" of the shallow lake, which is the area with the peak
contribution of our EC derived fluxes, especially during strong winds. The importance of fresh labile
organic matter provided by the die-back of the former vegetation as driving force for high $CH_4$
emissions was also discussed in Hahn et al. (2015). They measured the highest $CH_4$ emissions in sedge
stands suffering from strongest die-back.
For comparison annual budgets of $CO_2$ and $CH_4$ for other nutrient-rich lentic freshwater ecosystems in
terms of pristine, anthropogenically influenced and transient ecosystems are listed in Table 4. Studies
on nutrient-rich lakes generally revealed lower $CH_4$ release for open water. In contrast, beaver ponds
were partially reported to emit similar rates of $CH_4$. Similarly to our study site beaver ponds are at least
in the beginning disbalanced ecosystems due to a rapidly increased water level with associated
suffering and finally the die-back of former vegetation, which is not adapted to higher water levels. A
large C pool for $CH_4$ production develops. However, even for a beaver pond existing more than 30 years
$CH_4$ emissions still account for 40 g $CH_4$ $m^{-2}$ $a^{-1}$ (Yavitt et al. 1992).
Annual $CH_4$ emissions of the surface type emergent vegetation were about 4-fold lower than for open
water. This might be the result of increased $CH_4$ oxidation in the soil, as plants with aerenchymatic
tissue release oxygen into the rhizosphere, in reverse to the emission of $CH_4$ into the atmosphere
(Bhullar et al. 2013). Minke et al. (2015) highlight the difference in net $CH_4$ release for typical helophyte
stands with moderate emissions for *Typha* dominated sites. Besides the effect of the gas transport
within plants, lower water and sediment temperatures due to shading by the emergent vegetation
might yield lower $CH_4$ production than for open water. Furthermore, the soil of emergent vegetation
stands is generally only temporarily and partly inundated and the water table decreased additionally
during the unusual warm and dry summer 2013, probably resulting in a lower rate of anaerobic
decomposition to $CH_4$. This in turn might be a reason, that in comparison to other sites dominated by
*Typha* (rewetted wetlands, lake shores and freshwater marshes; see Table 4) the emergent vegetation
at our site is at the lower limit of reported $CH_4$ release rates and best comparable to closed chamber
measurements of *Typha latifolia* microsites at another rewetted fen site in NE Germany (Günther et
al. 2015).

## 4.3   Annual net $CO_2$ release

We observed high annual net release of $CO_2$ during the observation period, which is rather uncommon
for fens several years after rewetting (e.g. Hendriks et al. 2007, Schrier-Uijl et al. 2014, Knox et al.
2015). Surprisingly, net $CO_2$ budgets were similar to those of drained and degraded peatlands (e.g.
Hatala et al. 2012, Schrier-Uijl et al. 2014). Both surface types acted as net sources, with emergent



vegetation (750 g $CO_2$ $m^{-2}$ $a^{-1}$) showing a distinctively higher net budget as well as GPP and $R_{eco}$ rates
than open water (158 g $CO_2$ $m^{-2}$ $a^{-1}$). Only few NEE rates are published for the open water body of
eutrophic shallow lakes. Ducharme-Riel et al. (2015) report 224 g $CO_2$ $m^{-2}$ $a^{-1}$ as annual NEE of a
eutrophic lake in Canada (see Table 4). According to Kortelainen et al. (2006) Finnish lakes, which are
mainly small and shallow, continuously emit $CO_2$ during the ice-free period, positively correlated with
their trophic state.
Our study revealed a high annual net $CO_2$ release for emergent vegetation, which is in the wide range
of NEE rates for *Typha* sites reported in other studies, including both net $CO_2$ sources and sinks (see
Table 5). GPP and $R_{eco}$ are generally high (especially at rewetted fen sites; both component fluxes most
often > 3000 g $CO_2$ $m^{-2}$ $a^{-1}$), characterising *Typha* stands as high turnover sites, usually resulting in net
$CO_2$ uptake. In contrast, $R_{eco}$ and GPP rates at our study site are in the lower part of the reported range.
We assume the continuously high $R_{eco}$ rates during winter 2013/2014, contributing to the high annual
net $CO_2$ emissions, to be the result of mild and dry meteorological conditions. In summer 2013, $R_{eco}$
exceeded GPP already in late June, indicating a significant contribution of heterotrophic respiration to
the $CO_2$ production. We cannot completely exclude a misestimation of the $CO_2$ exchange during
midsummer due to longer data gaps. However, unusual warm and dry conditions and associated water
table lowering during summer 2013 might have triggered a shift from anaerobic to aerobic
decomposition. This includes the exposed organic mud at former shallowly inundated soil of emergent
vegetation stands, e.g. at the edge of the lake. Besides $CH_4$, Hahn-Schöfl et al. (2011) showed that the
new sediment layer at the bottom of inundated areas exhibits very high rates of anaerobic $CO_2$
production. The effect of water table lowering at *Typha* sites due to dry conditions is also shown by
Günther et al. (2015) and Chu et al. (2015): relative increase of $R_{eco}$ rates, resulting in net $CO_2$ release.
This might be of special interest in terms of climate change, as a temperature increase and significantly
less precipitation in summer are expected for NE Germany. In addition, a considerable increase of
microbial activity and thus, generally increased decomposition due to high temperatures might be of
importance. Allochthonous organic matter import into the NE bay due to lateral transport, as discussed
for $CH_4$, might have further enhanced decomposition (e.g. Chu et al. 2015).

## 4.4   Global warming potential and the impact of spatial heterogeneity

The lake ecosystem is characterised by a high GWP 9 years after rewetting, mainly driven by high $CH_4$
emissions. Based on our results the site can hardly be classified into any phase following peatland
rewetting discussed by Augustin and Joosten (2007). The slow development and shift of the ecosystem
to a C sink with reduced climate impact might be the result of the exceptional characteristics



represented by eutrophic conditions and lateral transport of organic matter within the open water
body. The trophic status of water and sediment is an important factor regulating GHG emissions, as
shown by Schrier-Uijl et al. (2011) for lakes and drainage ditches in wetlands. However, the unusual
meteorological conditions during our study period might have caused a comparable low GWP
compared to previous years due to lower $CH_4$ emissions at the expense of high net $CO_2$ release. In
comparison, e.g. Schrier-Uijl et al. (2014) report C uptake and a GHG sink function of a fen 10 years
after rewetting with water levels below or at the soil surface. In a study by Knox et al. (2015) a wetland
with mean water level above the soil surface was characterised by a near-neutral climate impact after
15 years of rewetting, where continued high $CH_4$ emissions were compensated by strong net $CO_2$
uptake. In the course of rewetting the water table is recommended to be held at or just below the soil
surface to prevent inundation and thus, the formation of organic mud (Couwenberg et al. 2011,
Joosten et al. 2012, Zak et al. 2015).
We calculated the "true" fluxes of $CO_2$ and $CH_4$ for the AOI by weighting the non-linear regression
functions for the two surface types with their fractional coverage inside the AOI. The inferred C budget
and global warming potential differs considerably from that of the EC source area, highlighting the
strikingly different emission rates of open water versus emergent vegetation. Thus, footprint analysis
providing the fractional coverage of the main surface types is imperative for the interpretation of
ecosystem flux measurements as provided by the EC technique at such a spatially heterogeneous site.
In addition, for an interannual comparison of EC derived budgets for such sites it is necessary to define
a fixed AOI, as the cumulative footprint climatology (representing the EC source area) changes
interannualy. Inter-site comparisons (e.g. with other shallow lakes evolved during fen rewetting) are
challenging with regard to the site-specific spatial heterogeneity.

## 5   Conclusions

This study contributes to the understanding of eutrophic shallow lakes as a challenging ecosystem
often evolving during fen rewetting in NE Germany. Within the study period the ecosystem was a
strong source of $CH_4$ and $CO_2$. Both open water and emergent vegetation, particularly including the
eulittoral zone, were net emitters of $CH_4$ and $CO_2$, but with strikingly different release rates. This
illustrates the importance of footprint analysis for the interpretation of the EC measurements on a
rewetted site with distinct spatial heterogeneity. Our results show that the intended effects of
rewetting in terms of $CO_2$ emission reduction and C sink recovery are not yet achieved at this site. The
negative climate impact of the lake is dominated by considerable $CH_4$ release, particularly from the
open water section. In combination with the high net $CO_2$ release the C budget seems to be extremely



unbalanced. Measurements of lateral transport of organic substrate within the open water body and a full C budget could give indication on a potential allochthonous input into the NE bay. Furthermore, the effect of unusual meteorological conditions need further investigation. A comparison with existing chamber measurements at the open water body for the same time period will be helpful for the evaluation of our measurements and estimation for the surface type fluxes. The site is continuously changing, with *Typha latifolia* progressively entering the open water body in the course of terrestrialisation, probably resulting in peat formation and C uptake once the shallow lake is replenished by organic sediments. Therefore, long-term measurements are necessary to evaluate the impact of future ecosystem development on GHG emissions. Moreover, statements for the climate impact of rewetted fens can only be provided by inclusion of additional sites varying in groundwater table and vegetation type. We assume that shallow lakes represent a special case with regard to the GHG dynamics and climate impact, with exceptionally high $CH_4$ release and occasionally high net $CO_2$ emissions. Inundation involves the risk of unpredictable and long-term high $CH_4$ emissions, especially in case of nutrient-rich conditions, that counteract the actually intended lowering of the climate impact of drained and degraded fens. We strongly recommend to consider this risk in future rewetting projects and support the call of Lamers et al. (2015) for the need of well-conceived restoration management instead of the trial-and-error approach, whereon restoration of wetland ecosystem services was based on for a long time.

## Acknowledgements

This work was supported by the Helmholtz Association of German Research Centres through a Helmholtz Young Investigators Group to T.S. (grant VH-NG-821) and is a contribution to the Helmholtz Climate Initiative REKLIM (Regional Climate Change). Infrastructure funding through the Terrestrial Environmental Observatories Network (TERENO) is also acknowledged. We thank M. Hoffmann (ZALF, Müncheberg, Germany) and C. Hohmann (GFZ Potsdam, Germany) for providing meteorological data.



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



Table 1: Data loss and final data coverage during the observation period. Percentage of $CO_2$ and $CH_4$
flux data lost by power and instrument failure and maintenance as well as quality control and footprint
analysis.

| Filter criteria | Percentage of data [%] | |
|---|---|---|
| | CO₂ | CH₄ |
| Power and instrument failure, maintenance | 15.0 | 46.4 |
| Absence of sensor | - | 11.2 |
| QC 2 | 7.5 | 2.0 |
| RSSI | - | 2.1 |
| u* | 18.6 | 8.8 |
| Unreasonably high fluxes | 0.2 | 0.1 |
| No footprint information/footprint > 20 % outside the AOI | 13.2 | 6.5 |
| **Final data coverage** | **45.5** | **22.9** |






Table 2: Gapfilling model performance was estimated according to Moffat et al. (2007) with several
measures ($n_{CO2}$ = 6193, $n_{CH4}$ = 3386, fluxes of best quality QC 0): the adjusted coefficient of
determination $R^2_{adj}$ for phase correlation (significant in all cases, p-value < $2.2e^{-16}$), the absolute root
mean square index ($RMSE_{abs}$) and the mean absolute error (MAE) for the magnitude and distribution
of individual errors, as well as the bias error (BE) for the bias of the annual sums.

| Method | $R^2_{adj}$ | $RMSE_{abs}$ (mg m$^{-2}$ 30min$^{-1}$) | MAE (mg m$^{-2}$ 30min$^{-1}$) | BE (g m$^{-2}$ a$^{-1}$) |
|---|---|---|---|---|
| $LUT_{CO2nofoot}$ | 0.74 | 104.35 | 24.05 | 13.14 |
| $NLR_{CO2foot}$ | 0.66 | 119.10 | 27.51 | -2.12 |
| $NLR_{CH4nofoot}$ | 0.79 | 1.36 | 0.83 | -3.34 |
| $NLR_{CH4foot}$ | 0.81 | 1.28 | 0.78 | -2.54 |




Table 3: Annual balances of $CO_2$ and $CH_4$ derived by different methods for the whole EC source area,
the area of interest (AOI) and the two surface types: LUT approach without footprint consideration
($LUT_{CO2nofoot}$), NLR approach without ($NLR_{CH4foot}$) and with ($NLR_{CH4foot}$, $NLR_{CO2foot}$) footprint
consideration. Uncertainty was calculated as square root of the sum of squared random uncertainty
(measurement uncertainty) and gapfilling uncertainty.

| Source area | Flux | Method | | | |
|---|---|---|---|---|---|
| | ($g\ m^{-2}\ a^{-1}$) | $CO_2$ | | $CH_4$ | |
| | | $LUT_{CO2nofoot}$ | $NLR_{CO2foot}$ | $NLR_{CH4nofoot}$ | $NLR_{CH4foot}$ |
| Whole EC source area | NEE | 524.5 ± 5.6 | 531.4 ±13.0 | | |
| | GPP | -2380.5 ± 5.6 | -2122.1 ± 16.7 | | |
| | $R_{eco}$ | 2863.6 ± 5.6 | 2603.6 ± 8.4 | | |
| | $CH_4$ | | | 40.5 ± 0.2 | 39.8 ± 0.2 |
| AOI | NEE | | 843.5 ±13.0 | | |
| | GPP | | -3192.2 ± 16.7 | | |
| | $R_{eco}$ | | 4035.7 ± 8.4 | | |
| | $CH_4$ | | | | 21.8 ± 0.2 |
| Emergent vegetation | NEE | | 750.3 ± 13.0 | | |
| | GPP | | -4076.8 ± 16.7 | | |
| | $R_{eco}$ | | 4827.2 ± 8.4 | | |
| | $CH_4$ | | | | 13.2 ± 0.2 |
| Open water | NEE | | 158.2 ± 13.0 | | |
| | GPP | | -1021.5 ± 16.7 | | |
| | $R_{eco}$ | | 1179.7 ± 8.4 | | |
| | $CH_4$ | | | | 52.6 ± 0.2 |




Table 4: NEE and net CH$_4$ exchange at open water sites. The letters in parentheses indicate seasonal
(S; May to October) and annual (A) budgets. Positive water level indicates inundated conditions. GHG
flux measurement methods are denoted as: CH = chambers, CO = concentration profiles, TR = gas traps.

| Reference | Location, ecosystem type | Dominant plant species | Study year | Average water depth (m) | NEE (g CO$_2$ m$^{-2}$ a$^{-1}$) | CH$_4$ (g CH$_4$ m$^{-2}$ a$^{-1}$) |
|---|---|---|---|---|---|---|
| Huttunen et al. (2003), CH | Lake Postilampi, Finland: hypertrophic lake | | 1997 | 3.2 | | 16 (A) |
| Casper et al. (2000), TR/CO | Priest Pot, UK: hypertrophic lake | | 1997 | 2.3 | | 13 (A) |
| Ducharme-Riel et al. (2015), CO | Bran-de-Scie, Quebec: eutrophic lake | | 2007-2008 | 3.2 | 224 (A) | |
| Wang et al. (2006), CH | Taihu Lake, China, hypertrophic lake: - bare infralittoral zone - pelagic zone | | 2003-2004 | 0.5 to 1.8 1.8 | | 3 (A) 4 (A) |
| Hendriks et al. (2007), CH | Horstermeer, The Netherlands: eutrophic ditches | | 2004-2006 | > 0 | | 5 (A) |
| Waddington and Day (2007), CH | Bois-des-Bel peatland, Quebec: - ponds - ditches | | 2000-2002 | > 0 > 0 | | 0.3 (S) 2.9 (S)[1] |
| Naimann et al. (1991), CH | Kabetogama Peninsula, Minnesota, beaver pond: - submergent aquatic plants - deep water | *Utricularia spp., Potamogeton spp.* | 1988 | 0.45 1.25 | | 14 (A) 12 (A) |
| Roulet et al. (1992), CH | Low forest region, Ontario: beaver ponds | | 1990 | 0.2 to 0.4 | | 7.6 (A) |
| Bubier et al. (1993), CH | Clay Belt, Ontario: beaver pond | | 1991 | 0.5 to 1.5 | | 44 (A) |
| Yavitt et al. (1992), CH | New York, beaver ponds: - 3 years old - > 30 years old | | 1990 | ≤ 2 ≤ 2 | | 34 (A) 40 (A) |



Table 5: Annual (A)/seasonal (S) NEE, GPP, $R_{eco}$ and net $CH_4$ exchange at *Typha* sites. Positive water
level indicates inundated soil. GHG flux measurement methods are denoted as: CH = chambers, EC =
eddy covariance.

| Reference | Location, ecosystem type | Dominant plant species | Study year | Mean water level (m) | NEE | GPP (g $CO_2$ m$^{-2}$ a$^{-1}$) | $R_{eco}$ | $CH_4$ (g $CH_4$ m$^{-2}$ a$^{-1}$) |
|---|---|---|---|---|---|---|---|---|
| Kankaala et al. (2004), CH | Lake Vesijärvi, Finland: - inner cattail-reed zone | *Phragmites australis, Typha latifolia* | 1997 | <0.1 to >0.2 | | | | 51 (S)[1] |
| | | | 1998 | <0.1 to >0.2 | | | | 43 (S)[1], 6 (S)[2] |
| | - outer cattail-reed zone | *Phragmites australis, Typha latifolia* | 1997 | <0.1 to >0.2 | | | | 30 (S)[1] |
| | | | 1998 | <0.1 to >0.2 | | | | 23 (S)[1], 7 (S)[2] |
| | | | 1999 | <0.1 to >0.2 | | | | 23 (S)[1] |
| Chu et al. (2015), EC | Lake Erie, Freshwater marsh | *Typha angustifolia, Nymphaea odorata* | 2011 | 0.3 to 0.6 | -289 (A) | -3338 (A) | 3049 (A) | 58 (A) |
| | | | 2012 | 0.3 to 0.6 | 109 (A) | -3490 (A) | 3599 (A) | 76 (A) |
| | | | 2013 | 0.3 to 0.6 | 340 (A) | -2666 (A) | 3006 (A) | 70 (A) |
| Bonneville et al. (2008), EC; Strachan et al. (2015), NEE: EC, CH4: CH | Mer Bleue, Canada, freshwater marsh | *Typha angustifolia* | 2005-2006 | winter > summer | -967 (A) | -3045 (A) | 2078 (A) | 170 (A) |
| | | | 2005-2009 | ≈ 0 | -462 to -1041 (A) | | | |
| Whiting and Chanton (2001), CH | Virginia, freshwater marsh | *Typha latifolia* | 1992-1993 | 0.05 to 0.2 | -3288 (A) | | | 109 (A) |
| | Florida, lake shore | *Typha latifolia* | 1992 | 0.05 to 0.2 | -3587 (A) | | | 69 (A) |
| | | | 1993 | 0.05 to 0.2 | -4177 (A) | | | 96 (A) |
| Rocha and Goulden (2008), EC | San Joaquin Freshwater Marsh Reserve, California: - freshwater marsh | *Typha latifolia* | 1999 | winter +, midsummer - | -929 (A) | -3994 (A) | 4811 (A) | |
| | | | 2000 | winter +, midsummer - | 1887 (A) | -6006 (A) | | |
| | | | 2001 | winter +, midsummer - | -1349 (A) | | 5980 (A) | |
| Knox et al. (2015), EC | - wetland (rewetted 2010) | *Schoenoplectus acutus, Typha spp.* | 2012 | 1.07 | | -7717 (A) | 6721 (A) | 71 (A) |
| | - wetland (rewetted 1997) | *Schoenoplectus acutus, Typha spp.* | 2012 | 0.26 | -1455 (A) | -5519 (A) | 4064 (A) | 52 (A) |
| Petrescu et al. (2015), EC | - wetland (rewetted 2010) | ? | 2010 | 0.51 | 388 (A) | | | 21 (A) |
| Minke et al. (2015), CH | Giel'čykaŭ Kašyl, Belarus, fen (rewetted 1985) | *Typha latifolia, Hydrocharis morsus-ranae* | 2010-2011 | 1 | 553 (A) | -2825 (A) | 3375 (A) | 80 (A) |
| | | | 2011-2012 | 0.7 | -414 (A) | -3980 (A) | 3566 (A) | 91 (A) |
| Günther et al. (2015), CH | Trebeltal, Germany, fen (rewetted 1997) | *Typha latifolia* | 2011 | 0.02 | -156 (A) | | | 13 (A) |
| | | | 2012 | -0.09 | 345 (A) | | | 4 (A) |
| Wilson et al. (2007, 2008), CH | Turraun, Ireland, cutover bog (rewetted 1991) | *Typha latifolia* | 2002 | 0.07 | 975 (A) | -3272 (A) | 4064 (A) | 39 (A) |
| | | | 2003 | 0.03 | 1653 (A) | -4357 (A) | 6010 (A) | 29 (A) |

[1] open water period
[2] winter



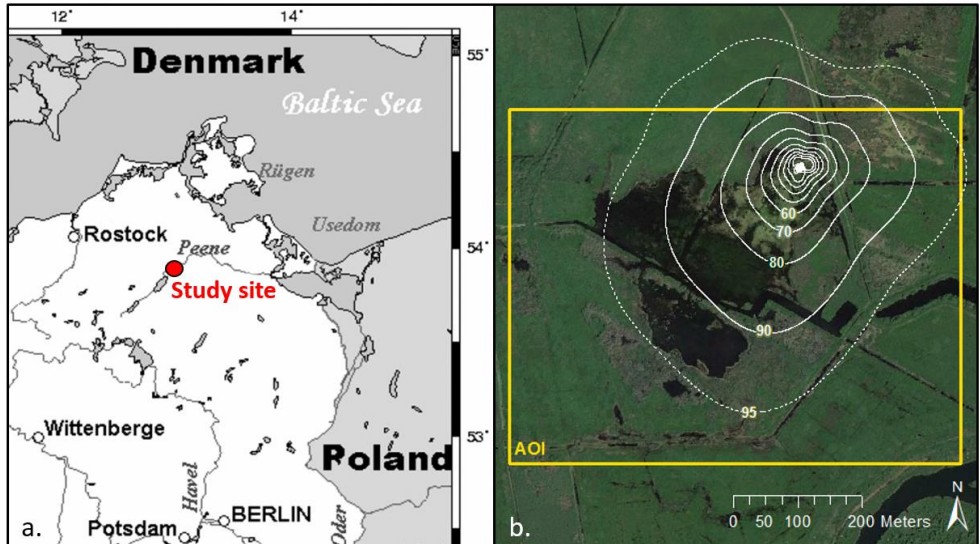


Figure 1: a) Polder Zarnekow is situated in NE Germany within the Peene River valley; map source and
copyright: https://commons.wikimedia.org/wiki/File:Germanymap2.png (modified). b) Footprint
climatology calculated according to Chen et al. (2011) on a Landsat image (6 Jun 2013, source: Google
Earth). White lines represent the isopleths of the cumulative annual footprint climatology, where the
area within the 95 isopleth indicates 95 % contribution to the annual flux. The white dot denotes the
tower position. The yellow box indicates the area of interest (AOI) as a filter criterion to focus on fluxes
of the shallow lake and to avoid the possible impact of a farm and grassland to the north of the shallow
lake. If the half-hourly flux source area exceeded the AOI by more than 20 % the flux was discarded.
The site is characterised by two main surface types: open water and emergent vegetation.





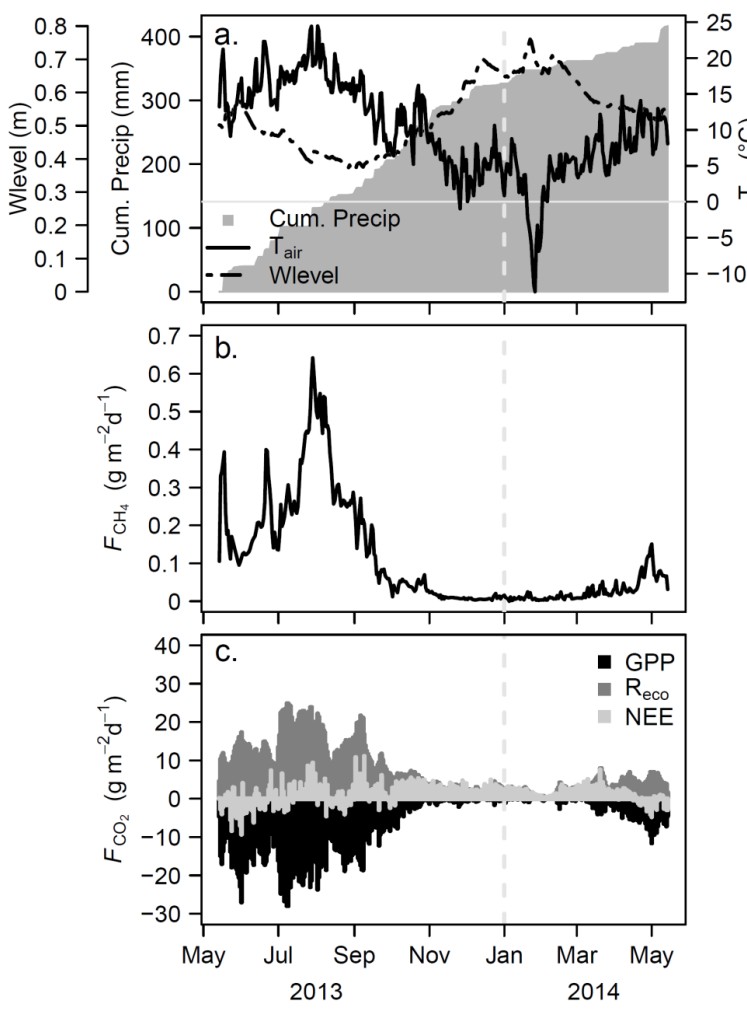


Figure 2: Temporal variability of environmental variables and ecosystem $CO_2$ and $CH_4$ exchange.
Seasonal course a) of water level (Wlevel), cumulative precipitation (Cum. Precip) and air temperature
($T_{air}$), b) the daily $CH_4$ flux (gapfilled, $NLR_{CH4nofoot}$) and c) the daily NEE (gapfilled $LUT_{CO2nofoot}$) and
component fluxes (modelled $R_{eco}$ and GPP, $LUT_{CO2nofoot}$).





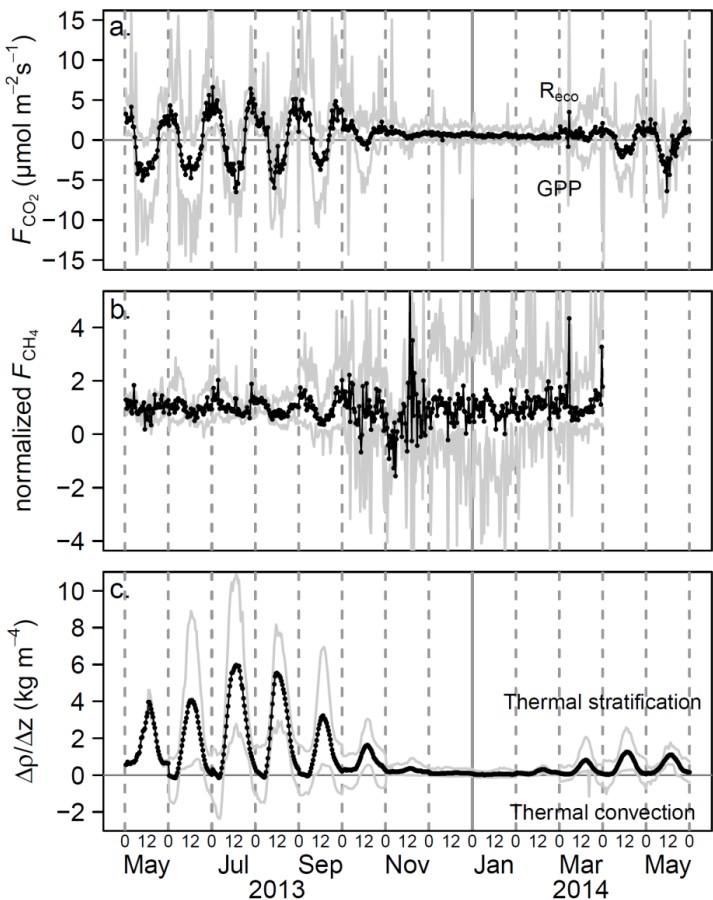


Figure 3: Average diurnal cycle of a) $CO_2$ flux, b) $CH_4$ flux and c) the water density gradient per month.

The numbers at the x-axis denote midnight (0) and midday (12). Midnight is also illustrated with a
dashed line. Black and grey lines represent the mean and the range, respectively. The $CH_4$ fluxes are
normalized with the monthly median of the half-hourly fluxes. Positive $CO_2$ fluxes represent the
dominance of $R_{eco}$ against GPP, negative fluxes the dominance of GPP against $R_{eco}$. The period of ice-
cover was excluded from the calculation of the temperature gradient. A density gradient equal to or
below zero indicates thermally induced convective mixing down to the bottom of the open water body
of the shallow lake, positive gradients instead thermal stratification.





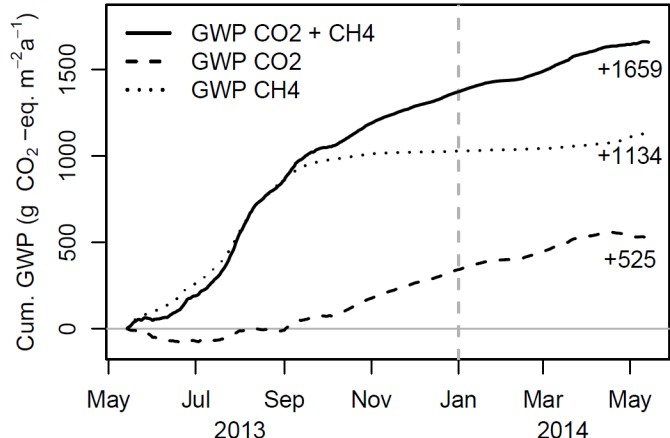


Figure 4: Cumulative GWP budgets of $CO_2$ (based on $LUT_{CO2nofoot}$), $CH_4$ (based on $NLR_{CH4nofoot}$) for the EC

source area and the sum of both during the observation period.




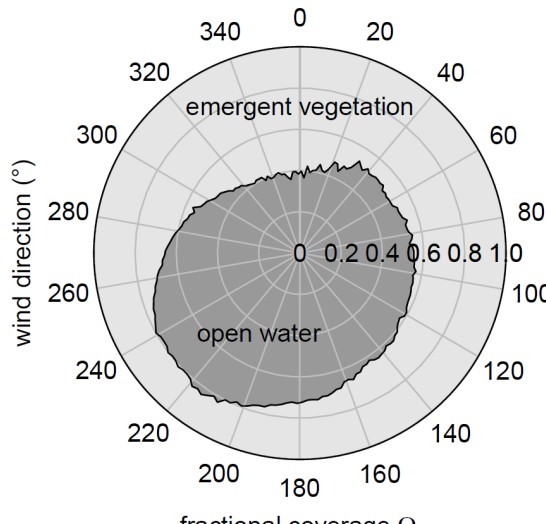


Figure 5: Source area fraction $\Omega_i$ of the two main surface types in dependence on the wind direction
(2°-bins).



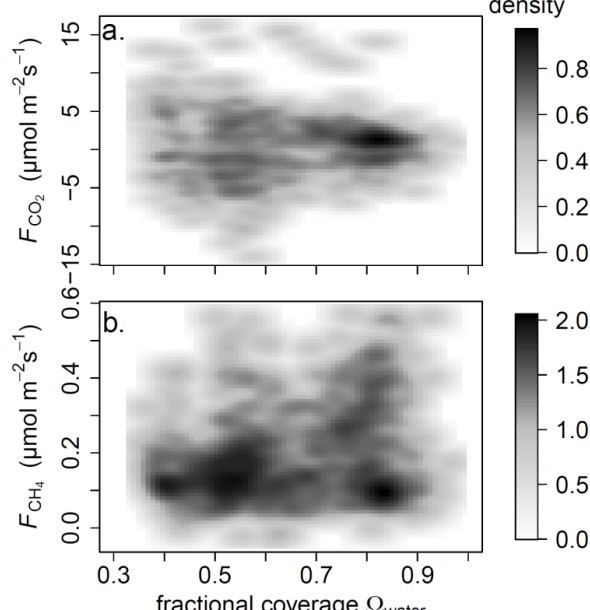


Figure 6: Impact of the fractional coverage of open water ($\Omega_{water}$) within the EC source area on the
measured fluxes of $CO_2$ and $CH_4$. The variability of $CO_2$ flux rates decreased with increasing $\Omega_{water}$,
whereas the variability of the $CH_4$ flux increased.



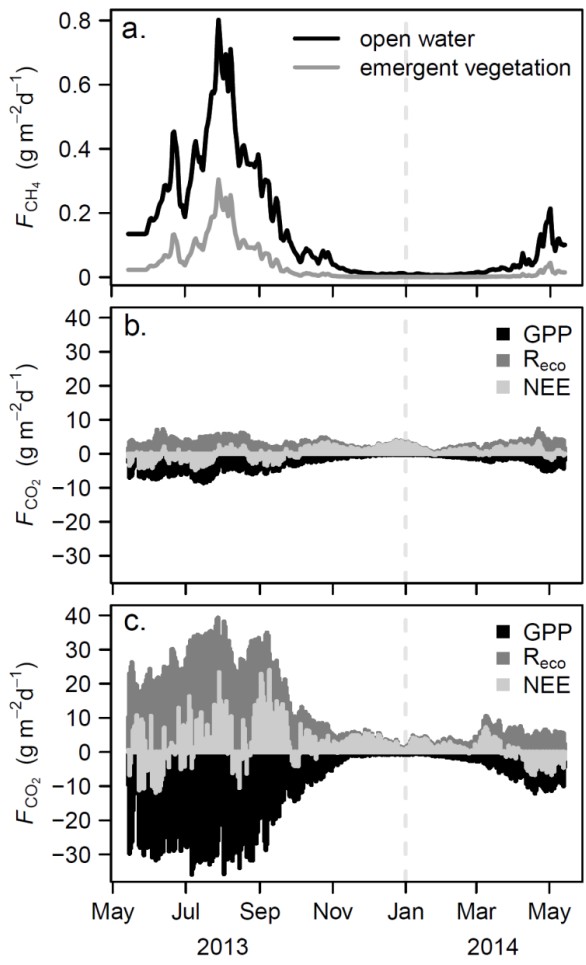


Figure 7: Daily CH$_4$, NEE and component fluxes (R$_{eco}$ and GPP) for the surface types: a) daily CH$_4$ flux of

open water and emergent vegetation, b) daily NEE and component fluxes for open water, c) daily NEE

and component fluxes for emergent vegetation, derived by NLR with the source area fractions of the

surface types ($\Omega_i$) as weighting factors (NLR$_{CH4foot}$, NLR$_{CO2foot}$).