# Peer review of "on a formerly drained fen"

_Biogeosciences, 2015_

## Referee Comment (RC1) · Anonymous Referee #1 · 17 Feb 2016

General This manuscript describes the effects of rewetting a fen in order to restore the function of fen as a carbon sink. The study site was monitored for a year with eddy covariance measurements of both CO2 and CH4 and subsequent calculations of CH4 and CO2 budgets, gross primary production, respiration, net ecosystem exchange and global warming potential of the gas fluxes. The study site showed a considerable carbon loss and global warming impact even after 9 years of rewetting. The study is well planned and the results presented mainly clearly. The scientific quality if good and the manuscript is well written. I think this is an important contribution to the scientific discussion, because the rewetting projects are widely planned and implemented. However, I have few suggestions for improvement of the manuscript.

1) In the manuscript there are lengthy descriptions of gap-filling of the eddy covariance data, and the coverage of the actual data is presented in Table 1. However, there is

very little information about the timing of these gaps, I was hoping for a bit more open policy about the shortcomings of the data. In row 310 there is a remark that data from April and May are missing from Figure 3 because the sensor was dismantled. Are there other similar longer gaps in the data? Where?

2) The term "polytrophic" is not very commonly used in the lake science, I suppose it means a shallow, polymictic and eutrophic lake. However, as the term is not very commonly known, I think the paper would draw more interest if the title was ". . .polymictic and eutrophic lake. . ." or ". . .a shallow eutrophic lake. . .".

3) The writers stated that summer 2013 was exceptionally hot and dry and as a consequence the water level dropped considerably rising again the next winter. As the lake is very shallow, I was wondering how much the fluctuation of the water level affected the lake are (i.e. area covered with water). Was the water area considerably larger in winter than in summer? One of the main findings of this study is that open water and vegetated areas had very different gas fluxes. How much did the fluctuating water level (or dry land versus water covered land) effect the results?

4) One of the findings of this study is that convection brought about a diurnal fluctuation of $CH_4$ flux. If this is true, most likely convection contributed also on the diurnal fluctuation of $CO_2$ flux. Have you considered this when calculating e.g. NEE?

Detailed comments:

Page 11, row 310: Please add 2014 to avoid misunderstandings (April and May 2014 not shown . . .)

Page 15, row 432: Extra bracket at the end of the sentence.

Figure 2. It is not quite clear here is the fluxes are for the whole EC area or for the AOI.

Figure 6. It is not quite clear what does the density describe. Please clarify.

---

## Referee Comment (RC2) · Anonymous Referee #2 · 25 Feb 2016

General Comments

This manuscript introduces a new data set on CO2 and CH4 eddy covariance fluxes above a formerly drained and recently rewetted fen area in northeastern Germany. The observations cover one year of flux data and show the area, part of which is permanently flooded, being a large greenhouse gas source. Since wetland restoration is an important and often controversial topic and flux measurements for this type of ecosystems are still scarce, the data set presented here does not only fit well into the scope of BG but has much relevance for the wider audience, too. In addition to this aspect, the study is innovative with respect to the spatial data analysis as it uses a footprint model to distinguish between the emissions from two different surface types using only one tower.

The data are well documented and apparently of high quality, and the paper is very well written – in fact far above average of first time submissions to this journal, as far as I have seen them! The manuscript has thus the potential to become a valuable (and probably much cited) contribution to BG. However, some sections of the Results, as well as parts of the Discussion, require some clarifications, and therefore I recommend that the authors be encouraged to carry out a (minor) revision that takes the following specific points into account.

Specific Comments

Line 218: What does 'enhanced' mean here – is this still simply a lookup table method or does it include something else?

Line 251: The outer pair of brackets is not needed here.

Line 300: The statements about the water level are confusing when comparing them with line 112 in the site description. There the water depth was said to 'range from 0.1 and 0.7 m' (does this refer to spatial or temporal variation?) and here the temporal fluctuations are shown to be 0.36 and 0.77 m as visible from Fig. 2. How do these two statements fit together?

Line 304: Why were median fluxes instead of averages or totals given here? I think this is not very common and should therefore be briefly explained.

Line 309ff: Why were the CH4 fluxes normalized but not the CO2 fluxes?

Line 363: Insert "for the AOI" before "than".

Lines 384ff: Would convection also affect the CO2 emissions from the lake? Please discuss whether this is possible – or why you think it's not.

Line 417: Replace "typically" with "typical".

Line 451: Add "and a higher rate of CH4 oxidation in the aerated top soil" after "CH4".

Lines 495ff: This is one of the (few) weak points of this study: With only one year of data that happened to be characterized by "unusual meteorological conditions" the question arises as to what extent the observation of the wetland being a large GHG source can be transferred to other sites and other years. Other studies have shown multi-year trends in GHG budgets following wetland restoration. I suggest that the authors discuss this in more detail, taking for example the papers by Waddington and Day (2007, JGR) or by Herbst et al. (2013, this journal) and/or the respective references therein into account.

Line 514: I suggest adding a phrase like "...and the interannual variability if short-term studies like this one are involved" to the end of this sentence.

Lines 517ff: What I miss in the conclusions is some statement or estimate that relates the finding of this study to the situation of drained fen grasslands, at least on the basis of literature data. Does the described method of rewetting (involving the flooding of substantial parts of the area) make the GHG budget worse than that of a drained fen? Or just worse than that of a more cautiously restored fen (with less surface inundation), but still better than that of the drained situation?

―――――――――――――――――――

---

## Author Comment (AC1) · 20 Apr 2016

**Author response to comments of Referee #1**

**(Biogeosciences Discuss., doi:10.5194/bg-2015-640-RC1, 2016)**

We thank Anonymous Referee #1 for the valuable and constructive comments, which helped us to further improve this manuscript. Below you will find the comments of Referee #1 followed by our responses which are marked in blue.

1. In the manuscript there are lengthy descriptions of gap-filling of the eddy covariance data, and the coverage of the actual data is presented in Table 1. However, there is very little information about the timing of these gaps, I was hoping for a bit more open policy about the shortcomings of the data. In row 310 there is a remark that data from April and May are missing from Figure 3 because the sensor was dismantled. Are there other similar longer gaps in the data? Where?

We added an appendix presenting the data coverage of $CO_2$ and $CH_4$ fluxes within the study period. We cross-refer to Fig. A1 on page 7 in line 185.

[Figure]

"Figure A1: Measurement coverage of a) $CO_2$ and b) $CH_4$ fluxes within the study period. Gapfilling results of the $MDS_{CO2nofoot}$ and $NLR_{CH4nofoot}$ approaches are added as grey circles."

2. The term "polytrophic" is not very commonly used in the lake science, I suppose it means a shallow, polymictic and eutrophic lake. However, as the term is not very commonly known, I think the paper would draw more interest if the title was "… polymictic and eutrophic lake …" or "… a shallow eutrophic lake …".

This comment is based on the very first submitted draft of this manuscript. However, this draft was slightly changed according to the quick reports of the Referees, which was necessary to publish the manuscript for interactive discussion. As suggested in the quick report, we changed the term "polytrophic" to "eutrophic" and thank Referee #1 for this suggestion. We now further replaced "eutrophic shallow lake" by "eutrophic and polymictic lake" in line 111 on page 4. In addition to the

suggestions in the quick reports, we applied few small changes to the very first draft to further improve the manuscript. Thus, the lines mentioned by Referee #1 are shifted.

3.  The writers stated that summer 2013 was exceptionally hot and dry and as a consequence the water level dropped considerably rising again the next winter. As the lake is very shallow, I was wondering how much the fluctuation of the water level affected the lake are (i.e. area covered with water). Was the water area considerably larger in winter than in summer? One of the main findings of this study is that open water and vegetated areas had very different gas fluxes. How much did the fluctuating water level (or dry land versus water covered land) effect the results?

During summer particularly areas with a wintertime very shallow inundation of the soil were exposed, pertaining especially parts of the emergent vegetation stands. We did not map the fluctuations of soil inundation and aerial images, which could help to define the extent of inundation, are not available for the periods with highest and lowest water table. Nevertheless, in summer the detection of inundated and exposed areas would be hampered by the vegetation hiding the surface. We could not observe a considerable decrease of the spatial extent of the open water body, as emergent vegetation mainly covers the shallower edges of the water body. Water table modelling would require a digital terrain model (DTM) with a very high height accuracy, as the study site itself is on average less than 0.5 m above sea level. The most accurate available DTM covering the site is the DTM5 with a height accuracy of 0.25 to 1 m, which is not sufficient to represent the microtopography.

Changing coverages of exposed versus inundated soil most probably have an effect on the difference of the surface type fluxes. However, for profound statements long-term measurements covering more than one summer will be necessary. In addition, we expect the effect of water level changes to be very variable within the open water body, as the bottom is characterised by a distinct microtopography (see also response to comment 3 of Referee #2) and therefore different vulnerability to changes. Thereby, eddy covariance measurements can only provide limited information.

We changed lines 476-479 on page 16 to the following: "Unusual warm and dry conditions and associated water table lowering during summer 2013 might have triggered a shift from anaerobic to aerobic decomposition due to the exposure of formerly only shallowly inundated soil and organic mud. However, this effect mainly concerns emergent vegetation stands. We could not observe a considerable decrease of the spatial extent of the open water body, as emergent vegetation mainly covers the shallower edges of the water body."

4.  One of the findings of this study is that convection brought about a diurnal fluctuation of $CH_4$ flux. If this is true, most likely convection contributed also on the diurnal fluctuation of $CO_2$ flux. Have you considered this when calculating e.g. NEE?

We did not consider convection within NEE modelling and the calculation of the surface type fluxes so far. However, we agree that thermally induced convective mixing might also have an effect on the diurnal fluctuations of NEE. Nevertheless, open water is characterised by remarkably lower $CO_2$ exchange rates than emergent vegetation.
According to our response we add the following paragraph to the discussion on the diurnal variability of $CH_4$ emissions (page 14, line 398): "Apart from $CH_4$, thermally induced convection potentially contributed also to the diurnal fluctuation of the $CO_2$ flux at our study site. According to Eugster et al. (2003) penetrative convection might be the dominant mechanism yielding $CO_2$ fluxes during periods of low wind speed, especially in case of a stratification of $CO_2$ concentrations in the water body. Ebullition triggered by convective mixing might be less important for $CO_2$ than for $CH_4$, as concentrations of $CO_2$ are most often low in gas bubbles (e.g. Casper et al. 2000, Poissant et al. 2007, Repo et al. 2007, Sepulveda-Jauregui et al. 2015, Spawn et al. 2015). Further investigations should

focus on the controls of the diurnal patterns in $CO_2$ and $CH_4$ exchange based on additional measurements, e.g. gas concentrations in the water, methane oxidation or plant-mediated transport."

*Detailed comments:*

5.  Page 11, row 310: Please add 2014 to avoid misunderstandings (April and May 2014 not shown …)

We changed the paragraph according to our response to comment Nr. 5 of Referee #2 and added the respective year to the months.

6.  Page 15, row 432: Extra bracket at the end of the sentence.

A cross-reference to Table 4 was missing. We already corrected this prior to the publication of the manuscript for interactive discussion as can be seen in line 435 on page 15 (for shifted lines see response to comment 2).

7.  Figure 2. It is not quite clear here is the fluxes are for the whole EC area or for the AOI.

Fig. 2 presents the daily fluxes for the EC source area. We added the missing information to the figure caption: "Figure 2: Temporal variability of environmental variables and ecosystem $CO_2$ and $CH_4$ exchange within the EC source area. Seasonal course a) of water level (Wlevel), cumulative precipitation (Cum. Precip) and air temperature ($T_{air}$), b) the daily $CH_4$ flux (gapfilled, $NLR_{CH4nofoot}$) and c) the daily NEE (gapfilled $LUT_{CO2nofoot}$) and component fluxes (modelled $R_{eco}$ and GPP, $LUT_{CO2nofoot}$)."

8.  Figure 6. It is not quite clear what does the density describe. Please clarify.

We thank the referee for this suggestion. We use a smoothed 2d kernel density estimate to illustrate the abundance of the $CO_2$ and $CH_4$ fluxes dependent on the fractional coverage of open water within the EC source area. The plot was created with the command smoothScatter of the R package graphics. The graph is based on flux data from 15 May till 14 September 2013, as the dependence of the flux variability on the source area coverage of open water is most pronounced during summer.
We changed the figure caption to the following: "Figure 6: Impact of the fractional coverage of open water ($\Omega_{water}$) within the EC source area on the measured fluxes of $CO_2$ and $CH_4$ (15 May to 14 September 2013). The abundances of $CO_2$ and $CH_4$ fluxes in dependence on $\Omega_{water}$ are illustrated by a smoothed two-dimensional kernel density estimate. The variability of $CO_2$ flux rates decreased with increasing $\Omega_{water}$, whereas the variability of the $CH_4$ flux increased."

---

## Author Comment (AC2) · 20 Apr 2016

In the following we add the additional reference information to our response. Note that we do not list references which are already mentioned in the Discussion Paper.

Poissant, L., Constant, P., Pilote, M., Canário, J., O'Driscoll, N., Ridal, J. and Lean, D.: The ebullition of hydrogen, carbon monoxide, methane, carbon dioxide and total gaseous mercury from the Cornwall Area of Concern, Sci. Total Envir., 381, 256-262, doi:10.1016/j.scitotenv.2007.03.029, 2007.

Sepulveda-Jauregui, A., Walter Anthony, K. M., Martinez-Cruz, K., Greene, S. and Thalasso, F.: Methane and carbon dioxide emissions from 40 lakes along a north-south latitudinal transect in Alaska, Biogeosciences, 12, 3197-3223, doi:10.5194/bg-12-3197-2015, 2015.

[Figure]

Spawn, S. A., Dunn, S. T., Fiske, G. J., Natali, S. M., Schade, J. D. and Zimov, N. S.: Summer methane ebullition from a headwater catchment in Northeastern Siberia, Inland Waters, 5, 224-230, doi:10.5268/IW-5.3.845, 2015.

---

## Author Comment (AC3) · 20 Apr 2016

**Author response to comments of Referee #2**

**(Biogeosciences Discuss., doi:10.5194/bg-2015-640-RC2, 2016)**

We are very grateful for the detailed and constructive comments provided by Anonymous Referee #2. They especially helped us to improve the results, discussion and conclusion parts of our manuscript. Below we listed the comments of Referee #2 followed by our responses which are marked in blue.

1. Line 218: What does 'enhanced' mean here – is this still simply a lookup table method or does it include something else?

Our "enhanced" Look-up Table (LUT) approach corresponds to the Marginal Distribution Sampling (MDS) approach (see e.g. Moffat et al. 2007). The term "enhanced" indicates an essential modification in comparison to the standard LUT: missing NEE is filled with the mean value of data under similar meteorological conditions (radiation, air temperature and vapour pressure deficit) of a fixed margin within a moving window. Thus, the temporal autocorrelation of NEE is exploited. The algorithm varies in case of incomplete meteorological data (see Reichstein et al. 2005). To adapt to the common terminology we replaced the abbreviation "LUT" by "MDS" at all occurences in the manuscript and changed page 8 lines 218-221 to: "A Marginal Distribtuion Sampling (MDS) approach proposed by Reichstein et al. (2005), available as web tool based on the R package REddyProc (http://www.bgc-jena.mpg.de/REddyProc/brew/REddyProc.rhtml) was applied for gapfilling and partitioning of NEE measurements (LUT$_{CO2nofoot}$), with air temperature as temperature variable."

2. Line 251: The outer pair of brackets is not needed here.

We agree and deleted the outer pair of brackets.

3. Line 300: The statements about the water level are confusing when comparing them with line 112 in the site description. There the water depth was said to 'range from 0.1 and 0.7 m' (does this refer to spatial or temporal variation?) and here the temporal fluctuations are shown to be 0.36 and 0.77 m as visible from Fig. 2. How do these two statements fit together?

We apologize for the confusion and the declaration of a rather misleading water level range. The range "0.1 to 0.7 m" on page 4 line 112 and page 7 line 206 is the generously rounded range of the mean annual water level 2008-2012 generated by measurement based water level modelling. For a long-term range we refer to Zak et al. (2015) reporting water levels between 0.2 m and 1.2 m above the surface at a specific gauge between 2004 and 2012. We replace the range "0.1 to 0.7 m" on page 4 line 112 and add in brackets "2004 to 2012; Zak et al. 2012". We deleted the water level information on page 7 line 206 as we declare the temporal range for our study period within the results part. This range is measured at one single position close to the tower, including the snow cover on ice covering the shallow lake. This measurement is not representative for the whole shallow lake, as the study site is characterised by a distinct microtopography due to previous shrinkage and subsidence of the peat in consequence of drainage and degradation.

4. Line 304: Why were median fluxes instead of averages or totals given here? I think this is not very common and should therefore be briefly explained.

We present median values for our flux measurements as this is the best measure of a central tendency in a skewed dataset due to not evenly distributed gaps.

5. Line 309ff: Why were the CH$_4$ fluxes normalized but not the CO$_2$ fluxes?

By normalising the mean half-hourly $CH_4$ fluxes per month we can illustrate the diurnal pattern of $CH_4$ fluxes, which was hardly visible in the unnormalised fluxes during months with generally low $CH_4$ exchange rates. We did not normalise the $CO_2$ fluxes so far as we can detect a diurnal cycle for the same months based on both normalised and unnormalised fluxes. However, to be consistent we now also normalised the mean half-hourly $CO_2$ fluxes per month. In addition, we decided to also include fluxes of days were less than five half-hourly flux values are available, thus including mean half-hourly $CH_4$ fluxes for April 2014, which are based on three days only, due to the dismantling of the sensor.

We modified lines 307-314 on page 11 to:
"To investigate the potential presence of a diurnal cycle of $CO_2$ and $CH_4$ fluxes throughout the study period we normalised the mean half-hourly $CO_2$ and $CH_4$ fluxes per month with the respective minimum/ maximum and median of the half-hourly fluxes of the specific month (modified from Rinne et al. 2007). A pronounced diurnal cycle of $CO_2$ fluxes with peak uptake around midday and peak release around midnight was obvious until November 2013 and beginning in March 2014 (see Fig. 3), although less pronounced in these two months. We found a clear diurnal cycle of $CH_4$ fluxes from June to September 2013 and in March 2014 (April 2014 based on 3 days only and May 2014 not available as the sensor was dismantled) with daily peaks during night-time (around midnight until early morning)."

We changed Fig. 3 as follows:

[Figure]

"Figure 3: Average diurnal cycle of a) $CO_2$ flux, b) $CH_4$ flux and c) the water density gradient per month. The numbers at the x-axis denote midnight (0) and midday (12) in UTC. Midnight is also illustrated with a dashed line. Black and grey lines represent the mean and the range, respectively. The $CO_2$ and $CH_4$ fluxes are normalised with the monthly minimum/ maximum and the median of the half-hourly fluxes, respectively. Although the zero line is slightly shifted due to normalisation, positive $CO_2$ fluxes roughly indicate the dominance of $R_{eco}$ against GPP, negative fluxes the dominance of GPP against $R_{eco}$. The period of ice-cover was excluded from the calculation of the temperature gradient. A density gradient equal to or below zero indicates thermally induced

convective mixing down to the bottom of the open water body of the shallow lake, positive gradients instead thermal stratification."

6. Line 363: Insert "for the AOI" before "than".

Done.

7. Lines 384ff: Would convection also affect the $CO_2$ emissions from the lake? Please discuss whether this is possible – or why you think it's not.

For our response to this comment we refer to our response to comment 4 of Referee #1.

8. Line 417: Replace "typically" with "typical".

Done.

9. Line 451: Add "and a higher rate of $CH_4$ oxidation in the aerated top soil" after "$CH_4$".

We agree and changed lines 448-451 on page 15, also considering the impact of soil shading: "Furthermore, soil shading potentially supports $CH_4$ oxidation, as the growth and activity of methanotrophic bacteria is reported to be inhibited by light (Dumestre et al. 1999, Murase and Sugimoto 2005). Besides, the soil of emergent vegetation stands is generally only temporarily and partly inundated and the water table decreased additionally during the unusual warm and dry summer 2013, probably resulting in a lower rate of anaerobic decomposition to $CH_4$ and a higher rate of $CH_4$ oxidation in the aerated top soil."

10. Lines 495ff: This is one of the (few) weak points of this study: With only one year of data that happened to be characterized by "unusual meteorological conditions" the question arises as to what extent the observation of the wetland being a large GHG source can be transferred to other sites and other years. Other studies have shown multi-year trends in GHG budgets following wetland restoration. I suggest that the authors discuss this in more detail, taking for example the papers by Waddington and Day (2007, JGR) or by Herbst et al. (2013, this journal) and/or the respective references therein into account.

The unusual meteorological conditions during our study period might have caused a differing GWP compared to years with usual meteorological conditions, highlighting the need of long-term measurements. Moreover, based on the few existing studies a consistent picture and development of the GHG exchange behaviour does not seem to exist for rewetted fens, probably due to a variety of driving conditions and processes. We agree to extend our comparison with other studies and for that refer to our response on comment 12 (changes for the paragraph of lines 495-504 on page 17).

In addition, we changed lines 491-494 on page 16f.: "Our results imply a delay of the ecosystem towards a C sink with reduced climate impact, which might be the result of the exceptional characteristics represented by eutrophic conditions and lateral transport of organic matter within the open water body."

Within the conclusions we deleted the sentence "Our results show […]" in lines 522f. and the sentences in lines 525-528 starting with "In combination with […]" and changed lines 534-536: "Inter-annual comparison are also necessary to verify what the results of this study imply: that the intended effects of rewetting in terms of $CO_2$ emission reduction and C sink recovery are not yet achieved at this site.

In this context, the effect of unusual meteorological conditions needs further investigation. More general statements for the climate impact of rewetted fens can only be provided by inclusion of additional sites varying e.g. in groundwater table and plant composition."

11. Line 514: I suggest adding a phrase like "… and the interannual variability if short-term studies like this one are involved" to the end of this sentence.

We agree and changed the sentence to: "Inter-site comparisons (e.g. with other shallow lakes evolved during fen rewetting) are challenging with regard to the site-specific spatial heterogeneity and further the interannual variability, if short-term studies like the present one are involved."

12. Lines 517ff: What I miss in the conclusions is some statement or estimate that relates the finding of this study to the situation of drained fen grasslands, at least on the basis of literature data. Does the described method of rewetting (involving the flooding of substantial parts of the area) make the GHG budget worse than that of a drained fen? Or just worse than that of a more cautiously restored fen (with less surface inundation), but still better than that of the drained situation?

The climate impact of our study site is stronger than generally expected for rewetted peatlands, apart from the $CH_4$ hot spot characteristic of newly rewetted sites. We mentioned in lines 459f. on page 15 that the net $CO_2$ budget for the EC source area at our study site was higher or similar to those of drained and degraded peatlands under grassland management (e.g. Hatala et al. 2012, Schrier-Uijl et al. 2014). In addition, $CH_4$ release was remarkably higher than for the referenced degraded sites, resulting in a stronger climate impact of our study site. Time plays an important role for the climate impact after rewetting and success is often achieved only several years or decades after rewetting (e.g. Hendriks et al. 2007/ Schrier-Uijl et al. 2014). Minke et al. (2015) showed still strong GHG emissions even after 25 years of rewetting due to strong above-surface water level fluctuations. However, the effect of water level does not seem to be consistent along different sites, especially for $CO_2$. Secondary plant succession towards a peat forming vegetation (Zerbe et al. 2013) and terrestrialisation (Zak et al. 2015) are reported to be requirements for peat formation and thus the revitalisation of the C sink function in case of inundated conditions in consequence of rewetting (but e.g. Knox et al. 2015, see response on comment Nr. 10). At our study site emergent vegetation, but especially non-peat-forming *Typha latifolia*, is progressively entering and organic mud is steadily filling up the open water body. Ongoing investigations will show, how the GHG exchange will develope.

We changed lines 400-409 on page 14 as follows and corrected a mistake in the emission factors derived from IPCC (2014):

"The $CH_4$ emissions of our studied ecosystem were within the range of other temperate fen sites rewetted for several years (up to 63 g $CH_4$ m$^{-2}$ a$^{-1}$; e.g. Hendriks et al. 2007, Wilson et al. 2008, Günther et al. 2013, Schrier-Uijl et al. 2014). This rate is remarkable higher than the emission factor of 28.8 g $CH_4$ m$^{-2}$ a$^{-1}$, that was assigned to rewetted temperate rich organic soils, which is in turn more than twice the rate of the nutrient-poor complement (IPCC 2014). In contrast, newly rewetted fens emit its multiple. In the first year after flooding, Hahn et al. (2015) observed an average net release of 260 g $CH_4$ m$^{-2}$ a$^{-1}$, which is 186 times higher than before flooding, at a fen site in NE Germany. Two years later the $CH_4$ emissions were considerably lower (40 g $CH_4$ m$^{-2}$ per growing season; Koebsch et al. 2015). However, natural (e.g. Bubier et al. 1993, Nilsson et al. 2001) and degraded fens (Hatala et al. 2012, Schrier-Uijl et al. 2014) release most often less $CH_4$ than the majority of rewetted fens, with some exceptions (e.g. Huttunen et al. 2003)."

In combination with comment Nr. 10 we changed the paragraph of lines 495-504 on page 17 as follows:

"However, the unusual meteorological conditions during our study period might have caused a differing (lower or higher) GWP compared to previous years. $CH_4$ emissions might have been lower at the expense of high net $CO_2$ release, whereas under usual meteorological conditions e.g. $CO_2$ uptake could probably compensate the $CH_4$ emissions. Inundation is generally associated with high $CH_4$ emission. Thus, the course of rewetting the water table is generally recommended to be held at or just below the soil surface to prevent inundation and thus, the formation of organic mud (Couwenberg et al. 2011, Joosten et al. 2012, Zak et al. 2015). In contrast to $CH_4$, the influence of water level on net $CO_2$ release is not consistent in the few existing studies of rewetted peatlands. In contrast to our site and e.g. Petrescu et al. (2015) and Minke et al. (2015), Knox et al. (2015) reported high net $CO_2$ uptake to substantially compensate high $CH_4$ emissions for a site with mean water levels above the soil surface after several years of rewetting (see Table 5). Similarly, Schrier-Uijl et al. (2014) reported high $CO_2$ uptake rates for a Dutch fen site 7 years after rewetting and even C uptake and a GHG sink function after 10 years with water levels below or at the soil surface. Herbst et al. (2011) present a snapshot of the GHG emissions of a Danish site after 5 years of rewetting with permanently and seasonally wet areas, whereby high $CO_2$ uptake and moderate $CH_4$ emissions lead to substantial GHG savings. In contrast, weak $CO_2$ uptake and decreasing, but still high $CH_4$ emissions were reported for another fen site in NE Germany with mean water levels above the soil surface (Koebsch et al. 2013, 2015 and Hahn et al. 2015), resulting in a decreasing climate impact after 3 years of rewetting. Interestingly, changes of NEE due to flooding were negligible, although GPP and Reco rates decreased considerable due to the flooding (Koebsch et al. 2013). In comparison to the decreasing CH4 emissions at this site, Waddington and Day (2007) report enhancing $CH_4$ release for a Canadian peatland in the first three years after rewetting. A third rewetted fen site in NE Germany with water levels close to the soil surface was reported as weak GHG source 14-15 years after rewetting (Günther et al. 2015)."

 We changed lines 538-540 as follows:

"Along with chamber measurements at the open water body our study shows that permanent (high) inundation in combination with nutrient-rich conditions involves the risk of long-term high $CH_4$ emissions. They counteract the actually intended lowering of the climate impact of drained and degraded fens and can result in an even stronger climate impact than degraded fens, as also shown by previous studies."

Apart from the suggestions of the two referees we decided to change lines 391-394 as follows: "Apart from convective mixing, highest sediment and soil temperature in the night till early morning might play an important role for the peak emissions of $CH_4$ due to increased microbial activity. Furthermore, diurnal variability in $CH_4$ oxidation could contribute to the daily pattern of $CH_4$ release. Oxygen is supplied to the water, sediment and soil during the day in consequence of photosynthesis and increases $CH_4$ oxidation. However, convective mixing of the water column during the night might supply oxygen to deeper water depths potentially increasing $CH_4$ oxidation. We assume plant-mediated transport to be characterised by a reverse diurnal cycle with peak emissions during day-time, as the release of methane is dependent on the stomatal conductance of the plants (e.g. Morrisey et al. 1993). This pathway is limited to plants with aerenchymatic tissue like *Typha latifolia*, which dominates the eulittoral zone at our study site. $CH_4$ is transported from the soil to the atmosphere, bypassing potential oxidation zones above the rhizosphere (chimney effect). Unusually for wetland plants (Torn and Chapin 1993), complete stomatal closure during night was observed for *Typha latifolia* (Chanton et al. 1993). However, this temporal constraint seems to be superimposed by more efficient $CH_4$ pathways during the night and early morning."

Furthermore, we added the following to line 514 on page 17: "Comparisons might be misleading in case the fractional coverages of the main surface types are not considered. Furthermore, as shown by

Wilson et al. (2007, 2008) and Minke et al. (2015) vegetation composition has a remarkable effect on GHG emissions of rewetted peatlands and should be considered in inter-site comparisons."

In addition, we recognized a mistake in Table 4 due to the erroneous line 7 ($CH_4$ emission from water) in Table 10 in Hendriks et al. (2007). In alignment with Table 7 in Hendriks et al. (2007) the right value for $CH_4$ emission from water for 2005 has to be 37.3 g C $m^{-2}$ $a^{-1}$, i.e. 46 g $CH_4$ $m^{-2}$ $a^{-1}$. We corrected the wrong value in Table 4 and also changed the study year of this observation to "2005". In addition, we added the annual net $CH_4$ exchange for 2006 according to Table 7 in Hendriks et al. (2007): 49 g $CH_4$ $m^{-2}$ $a^{-1}$ at water levels above 0 m.

**Additional references**

Note that we do not list references, which are already mentioned in the Discussion Paper.

Dumestre, J. F., Guézennec, J., Galy-Lacaux, C., Delmas, R., Richard, S. and Labroue, L.: Influence of light intensity on methanotrophic bacterial activity in Petit Saut Reservoir, French Guiana, Appl. Environ. Microb., 65, 534-539, 1999.

Koebsch, F., S.Glatzel, J.Hofmann, I. Forbrich, andG. Jurasinski: $CO_2$ exchange of a temperate fen during the conversion from moderately rewetting to flooding, J. Geophys. Res.-Biogeo., 118, 940-950, doi:10.1002/jgrg.20069, 2013.

Moffat, A. M., Papale, D., Reichstein, M., Hollinger, D. Y., Richardson, A. D., Barr, A. G., Beckstein, C., Braswell, B. H., Churkina, G., Desai, A. R., Falge, E., Gove, J. H., Heimann, M., Hui, D., Jarvis, A. J., Kattge, J., Noormets, A. and Stauch, V. J.: Comprehensive comparison of gap-filling techniques for eddy covariance net carbon fluxes, Agricultural and Forest Meteorology, 147, 209-232, doi:10.1016/j.agrformet.2007.08.011, 2007.

Murase, J. and Sugimoto, A.: Inhibitory effect of light on methane oxidation in the pelagic water column of a mesotrophic lake (Lake Biwa, Japan), Limnol. Oceanogr., 50, 1339-1343.